# Phase-selective recrystallization makes eutectic high-entropy alloys ultra-ductile

Qingfeng Wu[1,4], Feng He[1,4], Junjie Li[1], Hyoung Seop Kim [2,3] ✉,
Zhijun Wang [1] ✉ & Jincheng Wang [1] ✉

Excellent ductility is crucial not only for shaping but also for strengthening metals and alloys. The ever most widely used eutectic alloys are suffering from the limited ductility and losing competitiveness among advanced structural materials. Here we report a distinctive concept of phase-selective recrystallization to overcome this challenge for eutectic alloys by triggering the strain hardening capacity of the duplex phases completely. We manipulate the strain partitioning behavior of the two phases in a eutectic high-entropy alloy (EHEA) to obtain the phase-selectively recrystallized microstructure with a fully recrystallized soft phase embedded in the skeleton of a hard phase. The resulting microstructure fully releases the strain hardening capacity in EHEA by eliminating the weak boundaries. Our phase-selectively recrystallized EHEA achieves a high ductility of ~35% uniform elongation with true stress of ~2 GPa. This concept is universal for various duplex alloys with soft and hard phases and opens new frontiers for traditional eutectic alloys as high-strength metallic materials.

Eutectic alloys have achieved a dominant position in the history of human civilization[1,2], e.g., cast irons in agricultural society[3], casting aluminum alloys in modern industry[4], and eutectic high-entropy alloys (EHEAs) in advanced metallic materials[5–8]. Excellent castability, free of segregations/defects, and self-generated dual phases make eutectic alloys significantly advantaged in low-cost mass-production and balanced strength-ductility combination[9,10]. However, these advantages are fading away with the rapid development of advanced structural materials[11–15] due to the unsatisfactory deformability of eutectic alloys and limited metallurgical mechanisms to ductilize them.

The cracking of weak interfaces, including phase boundaries (PBs) and grain boundaries (GBs) of the hard phases, causes premature failure during the uniaxial tension of eutectic alloys[16,17]. This situation results in the low tensile elongation of eutectic alloys even though their phases are both strain hardenable intrinsically. Delaying crack initiation and preventing crack propagation are the only successful routes to sustain the strain hardening of eutectic alloys to date: for example,

refining the microstructures by controlling the solidification process[18,19] and recrystallizing the two phases through hot forming or severe plastic deformation followed by subsequent annealing[20,21]. Although these methods delay the early fracture of eutectic alloys, the strain hardenability of the eutectic phases is not fully triggered due to extensive crack nucleation, leaving it still challenging to obtain ultra-ductile bulk eutectic alloys.

In this work, we show that phase-selective recrystallization (PSR) achieves ultra-ductile eutectic alloys by taking full advantage of the strain hardening capacity of both phases in eutectics. Different from traditional strategies, we primarily focus on eliminating and confining the crack nucleation sites during deformation by tailoring the recrystallization behaviors of the constituent phases. With reduced and confined early cracks, the excellent strain hardening capacity of both phases in eutectics is completely released, rendering a twofold tensile elongation compared to those of As-cast (AC) and fully recrystallized (FR) eutectic alloys.

[1]State Key Laboratory of Solidification Processing, Northwestern Polytechnical University, Xi'an 710072, China. [2]Graduate Institute of Ferrous & Energy Materials Technology, Pohang University of Science and Technology (POSTECH), Pohang 37673, South Korea. [3]Advanced Institute for Materials Research (WPI-AIMR), Tohoku University, Sendai 980-8577, Japan. [4]These authors contributed equally: Qingfeng Wu, Feng He. ✉e-mail: hskim@postech.ac.kr; zhjwang@nwpu.edu.cn; jchwang@nwpu.edu.cn

## Results and discussion

### Microstructures and mechanical properties

We present PSR in a model EHEA $Ni_{30}Co_{30}Cr_{10}Fe_{10}Al_{18}W_2$ (in atomic percentage) consisting of face-centered cubic (FCC) and ordered body-centered cubic (B2) phases[22,23] in the current work. In the AC state, the EHEA exhibits a typical eutectic-dendritic structure, where the dendrite stems are well-aligned lamellar structures while the interlamellar regions are irregular duplex structures (Fig. 1a). Traditional recrystallization treatment of the FR alloys eliminates the lamellar structures, replaced by the equiaxed duplex structures with randomly distributed orientations (Fig. 1b). As reported, the lamellar structures can be retained in the FR alloys at very low annealing temperatures, resulting in the ultrafine-grained (UFG) EHEAs[7,24]. The PSR brings a distinct microstructure compared with the AC and FR alloys. The FCC phase shows approximately equiaxed grains with random orientations, while the B2 phase exhibits a skeletal morphology with several specific orientations (Fig. 1c), i.e., PSR. The increased frequency of twin boundaries (TBs) in the FCC phase and low angle GBs in the B2 phase confirms the separated recrystallization and recovery of the two phases (Supplementary Fig. 1). After PSR, the interfaces between the FCC and B2 phases deviate from the original Kurdjumove-Sachs (K–S) orientation relationship[25], and the sizes of both the two phases increase slightly (Supplementary Fig. 2).

Accompanied by PSR, the ductility of EHEAs increases significantly. As shown in Fig. 1d, the tensile true stress-strain curves of the AC, FR, and PSR EHEAs do not show any differences in the yielding behavior but exhibit remarkable differences in the fracture stress and strain. The AC EHEA only has a fracture true strain of ~14%. The PSR EHEA with a well-tailored PSR microstructure doubles the fracture strain to ~30%. In comparison, the classical FR EHEA only increases the fracture strain slightly to ~17%. More importantly, the PSR EHEA exhibits a tremendous strain hardening capacity until fracture, leading to the high fracture true stress of ~1850 MPa, much higher than those of the AC and FR alloys, ~1390 and ~1460 MPa, respectively.

The excellent ductility and strain hardenability provide great potential to further improve the strength. By introducing dislocations and precipitations, the engineering ultimate tensile strength of the further strengthened PSR EHEAs was tuned between ~1.8 and ~2.2 GPa (Supplementary Fig. 3), which is comparable to advanced high-strength metallic materials. In Fig. 1e, we summarized the tensile properties of the present PSR EHEA and further strengthened PSR EHEAs together with AC, FR, and UFG EHEAs[7,24,26–33]. The uniform elongation of the PSR EHEA (~33%) exceeds the maximum value (~25%) achieved by traditional thermomechanical treatments, while the engineering ultimate tensile strength approaches the strongest UFG EHEA. The further strengthened PSR EHEAs show great

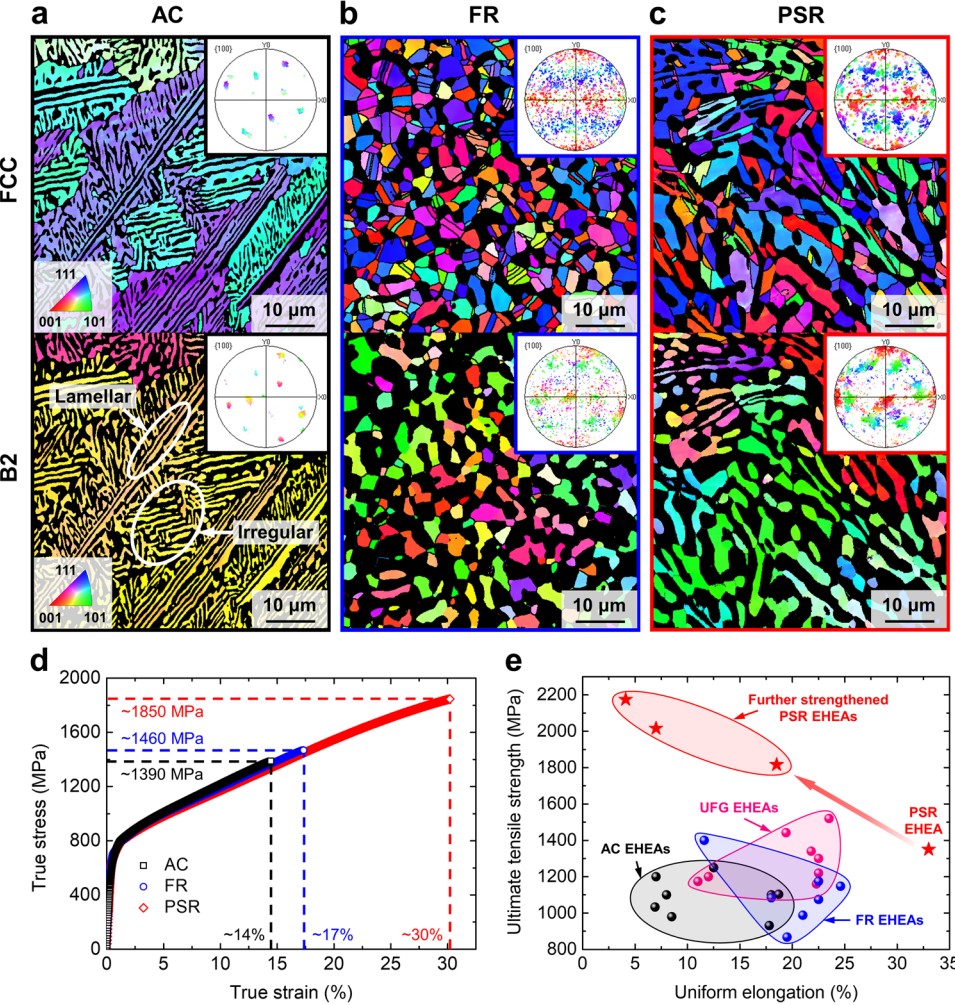

**Fig. 1 | Microstructures and mechanical properties of the PSR EHEA.**
**a**–**c** Electron backscattering diffraction (EBSD) inverse pole figure (IPF) maps of the FCC (upper row) and B2 (lower row) phases in the AC, FR, and PSR EHEAs, respectively. The insets show the corresponding pole figure (PF) maps. **d** Tensile true stress-strain curves of the AC, FR, and PSR EHEAs. **e** Ultimate tensile strength versus uniform elongation of the present PSR and further strengthened PSR EHEAs compared with traditional AC, FR, and UFG EHEAs[7,24,26–33].

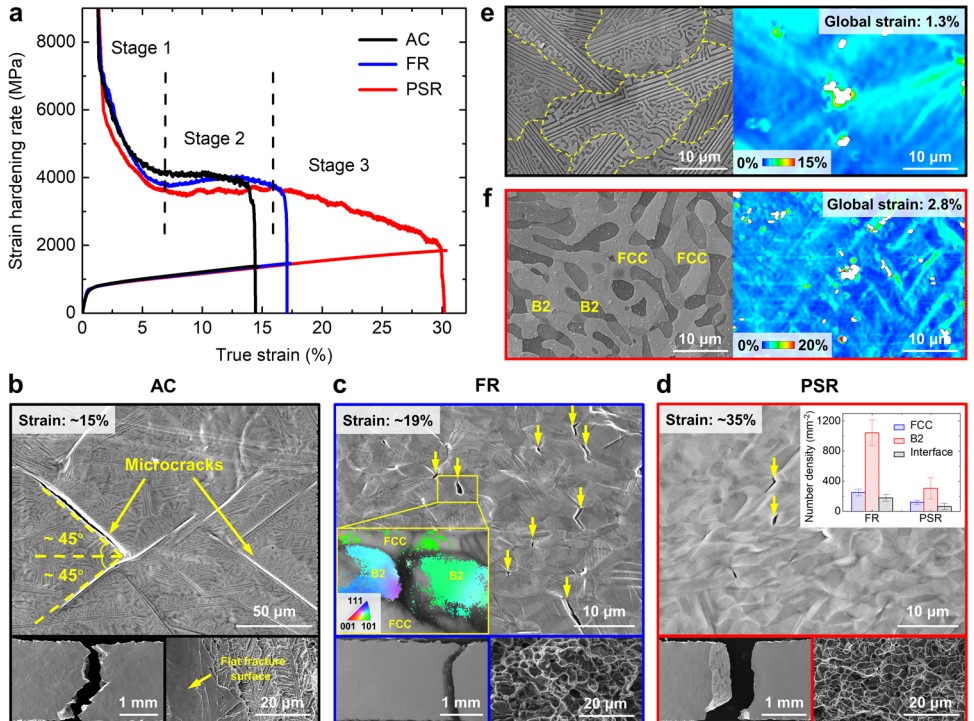

**Fig. 2 | Fracture mechanisms of the PSR EHEA. a** Strain hardening rate curves of the AC, FR, and PSR EHEAs. **b–d** Fracture cross-sections and surfaces of the AC, FR, and PSR EHEAs. The microcracks within the B2 phase are marked by the yellow arrows. The inset in **c** shows the EBSD IPF map of the B2 phase around a typical microcrack, revealing the crack initiation at the GB. The inset in **d** compares the number density of microcracks between FR and PSR EHEAs, revealing the decreased cracking propensity of the PSR EHEA. Error bars represent standard deviation. **e, f** μ-DIC results of the AC and PSR EHEAs during tensile deformation, respectively, revealing the severe strain localization in the AC EHEA and uniform strain distribution in the PSR EHEA. The tensile direction is horizontal in all images.

improvement in tensile strength compared with the traditional UFG EHEAs in previous reports[7,24].

## Fracture and deformation mechanisms

The significant increase in ductility of the PSR EHEA calls for a deeper understanding of the relationship between microstructures and deformation mechanisms. We firstly examined the strain hardening rate curves to understand the deformation behaviors at the macroscopic level (Fig. 2a). The PSR alloy exhibits a three-stage strain hardening behavior, i.e., the curve drops rapidly after yielding (Stage 1), then becomes stable gradually (Stage 2), and finally decreases slowly (Stage 3) until reaching the tensile plastic instability criterion[34]. However, the AC and FR alloys only have a two-stage strain hardening behavior and fracture before meeting the instability criterion. The premature failures of the AC and FR alloys are responsible for their relatively low ductility. In comparison, the PSR alloy exhausts its strain hardening capacity before facture, which is unusual in eutectic alloys.

We attribute these distinct strain hardening behaviors of the AC, FR, and PSR EHEAs to their entirely different microstructures (Fig. 1a–c) based on the close examination of the fracture and deformation mechanisms. Figure 2b–d presents the fracture cross-sections and surfaces of the AC, FR, and PSR EHEAs during in situ tensile tests to distinguish the different fracture mechanisms. On the cross-section of the fractured AC specimen, a large number of microcracks appear at the PBs in the lamellar regions, which have an angle of ~45° with the tensile direction (Fig. 2b and Supplementary Fig. 4). The connection of these microcracks results in the zig-zag morphologies of the main crack and partially flat fracture surface. Therefore, the premature failure of the AC EHEA is caused by the decohesion of the lamellar PBs with specific orientations. After eliminating the lamellar structures by recrystallization, the primary crack mainly origins from the GBs of the B2 phase (indicated by the yellow arrows), and the fracture surfaces

exhibit typical dimple morphologies in the FR and PSR EHEAs (Fig. 2c, d). As revealed by the in situ tensile tests (Supplementary Figs. 5, 6), the microcracks occur at the GBs of B2 at ~17% nominal strain, which is consistent with the fracture strain of the FR alloy, further proving its GB cracks-induced premature failure. However, the PSR alloy shows amazing crack resistance. The microcrack density of the fractured PSR specimen is only one-third of the FR specimen, even under a twofold nominal strain (Fig. 2d). This is because the B2 phase in the PSR alloy is an integrated skeleton while that in the FR alloy is recrystallized with GBs. The GB density of B2 in the PSR EHEA is much lower than that in the FR EHEA due to the non-recrystallization state after PSR. Therefore, from the perspective of fracture, the PSR prevents premature failure and enhances ductility by eliminating the lamellar structures in the AC alloy while avoiding the high-density PBs of the B2 phase in the FR alloy.

We conducted in situ microscopic-digital image correlation (μ-DIC) experiments to reveal the underlying reasons for the different fracture modes. In the AC state, the grains having 45° oriented lamellar structures with the tensile direction exhibit severe strain localization (Fig. 2e) and become crack initiations. Such an orientation-dependent strain localization behavior has also been observed in lamellar-structured TiAl alloys[35] and pearlitic steels[36], and can be well interpreted by dislocation pile-up model. After PSR, the grain-scale strain localization is released, and strain partitioning only exists between the two phases of FCC and B2 (Fig. 3f), thereby preventing the crack nucleation[37,38]. In this case, the weak GBs of the load-bearing B2 phase become the crack initiations in the FR and PSR alloys.

Apart from these distinct fracture mechanisms, the PSR also stimulates extra deformation mechanisms, which assure the third stage of strain hardening in the PSR EHEA (Fig. 2a). To validate this standpoint, we conducted detailed transmission electron microscope (TEM) analyses. As shown in Fig. 3a, dense plane slip bands are characterized

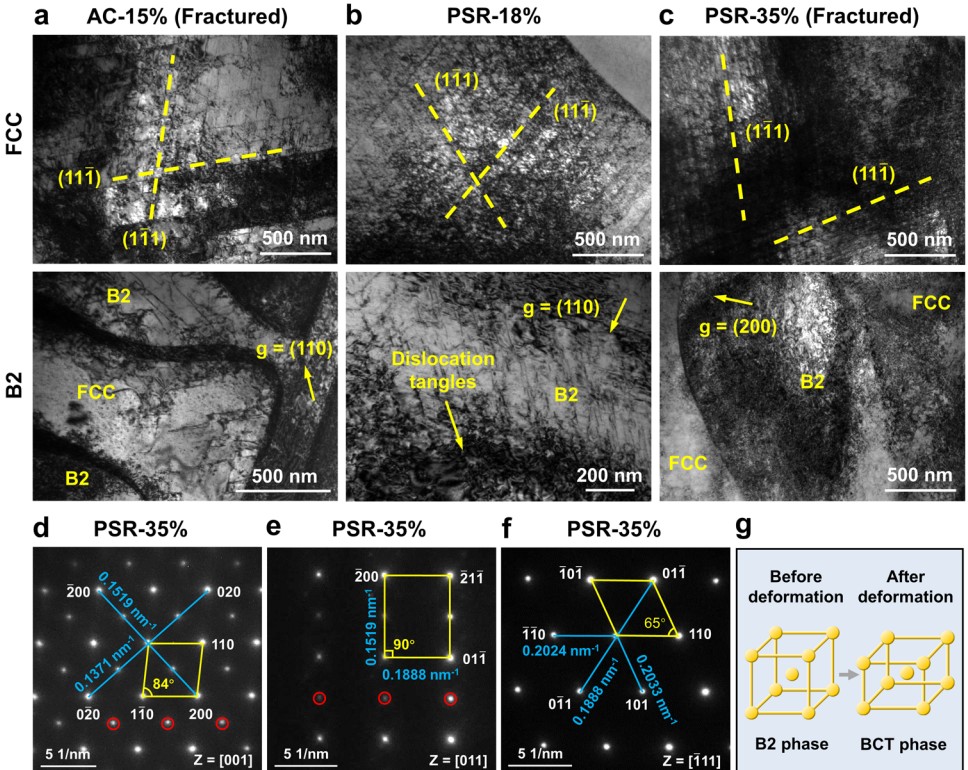

**Fig. 3 | Deformation mechanisms of the PSR EHEA. a** Dislocation substructures in the fractured AC EHEA. **b**, **c** Dislocation substructures in the 18% and 35% stretched PSR EHEAs, respectively. **d**–**f** SAED patterns of the original B2 phase in the fractured PSR specimen taken from the zone axis of [001], [011], [Ī11], respectively, revealing the body-centered tetragonal (BCT) crystal structure. The superlattices are marked by the red circles. **g** Schematic diagram illustrating the crystal structure transformation before and after deformation.

in the FCC phase of the fractured AC alloy, while much less dislocations are detected in the B2 phase with 15% nominal strain. Moreover, dislocations in the B2 phase tend to appear near the PBs. In the PSR alloy with a similar nominal strain (18%, Fig. 3b), however, both the FCC and B2 phases exhibit a high density of dislocations. We also examined the dislocation structures of the PSR EHEA at nominal strains of 0.5% and 8%, respectively (Supplementary Fig. 7). Well-developed planar slip bands in both phases show that the FCC and B2 phases in the PSR EHEA contribute to the strain hardening simultaneously from the onset of plastic deformation. As the strain further increases, the dislocation density significantly increases in both phases (Fig. 3c), illustrating their superior dislocation storage capacity. Especially for the B2 phase, even though no evidence for slip transfer across the PBs[39,40] is detected, the naturally superior deformability allows a high density of dislocations to be activated. As a result, the geometrically necessary dislocations significantly multiply due to the heterogeneous microstructure, promoting the sustainable increase of back stress strengthening[41,42]. By conducting the loading-unloading-reloading experiments, the back stress is quantitatively measured to be ~900 MPa near failure (Supplementary Fig. 8), accounting for the high strain hardening capacity of the PSR EHEA.

More importantly, a phase transformation in the B2 phase is activated by the increased strain. We observed a B2 → body-centered tetragonal (BCT) phase transformation in the PSR EHEA during deformation. The selected area electron diffraction (SAED) patterns of the B2 phase taken from the [001] zone axis transfer from square-arranged spots before deformation to parallelogram-arranged spots after deformation (Fig. 3d), indicating the distinct interplanar spacings between (200) and (020) planes. Quantitative calculations (Fig. 3d–f and Supplementary Fig. 9) reveal the crystal structure expands ~6% along the (200) direction while decreases ~5% along the (002) and

(020) directions after deformation, as schematically illustrated in Fig. 3g. The ductile-transformable B2 phase[43] has been proven to enhance the fatigue life significantly in the duplex HEA[44]. In the present PSR EHEA, the phase transformation enhances the deformability of the B2 phase and promotes the coordinated deformation between the soft and hard phases, even with low content of transformation. Therefore, the sustainable strain hardening of the PSR EHEA beyond 18% strain is accomplished by activating high-density dislocations in the FCC and B2 phases (Fig. 3c) and phase transformation in the B2 phase (Fig. 3g).

## PSR mechanisms

We have identified the great advantages of PSR on improving the mechanical properties and revealed the unique mechanisms. The main discrepancy between traditional FR EHEAs and the present PSR EHEA is the non-recrystallization state and skeletal morphology of the B2 phase. The key of PSR originates from our critical thinking on tailoring the strain partitioning behaviors in duplex alloys: that is, moderate deformation will induce two different levels of strain in the two phases, where one is for recrystallization while the other can only cause recovery during subsequent thermal treatment. The recovery and recrystallization behaviors of the FCC and B2 phases in the EHEA under different deformation amounts and subsequent annealing were investigated experimentally in Supplementary Fig. 10. Apparently, 30% reduction is a suitable deformation amount to achieve partial recrystallization in FCC while only recovery in B2. Accordingly, the unique processing routes developed for PSR include two cycles of 30% moderate deformation and annealing (Fig. 4a). After one cycle of thermomechanical treatment, the FCC phase in the irregular regions recrystallizes partially (Supplementary Fig. 11a–c). After two cycles of thermomechanical treatment, the FCC phase recrystallizes completely while the B2 phase recoveries and coarsens.

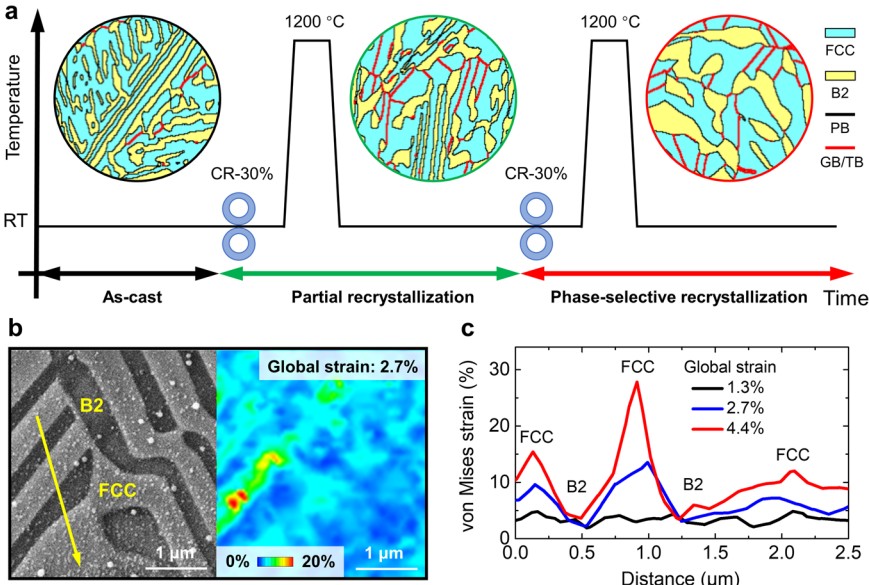

**Fig. 4 | Processing routes and formation mechanisms of the PSR. a** Schematic illustration of the processing routes and corresponding microstructure evolution. For PSR, the AC EHEA was cold-rolled (CR) 30% and annealed at 1200 °C for two cycles. The FCC phase, B2 phase, PB, and GB/TB are marked in cyan, yellow, black, and red colors, respectively. **b** In situ μ-DIC results of the AC EHEA during tensile test, revealing the strain partitioning between the FCC and B2 phases during deformation. **c** Variation of the local von Mises strain along the yellow arrow in **b**, revealing the much higher strain partitioned to the FCC phase.

In view of the critical role of the strain partitioning between the two phases in PSR, we further uncovered the stran partitioning behaviors. Since the microhardness of the FCC phase (~4.44 GPa) is much lower than the B2 phase (~5.67 GPa) (Supplementary Fig. 11d–f), it will bear more strain during deformation. In situ μ-DIC technique was used to characterize the strain partitioning behaviors experimentally. A distinct strain difference exists between the two phases during tensile deformation (Fig. 4b). Quantitative analysis reveals that the local von Mises strain in the FCC phase can be even up to 6 times higher than that in the B2 phase (Fig. 4c). Therefore, it is reasonable that cold rolling will induce more strain energy in the FCC phase, which promotes the subsequent individual recrystallization[45,46].

Theoretically, the strain partitioning in the constituent phases during deformation can be described as the functions related to the strength and work hardening parameters of the eutectic alloy and its constituent phases, and the volume fraction of the constituent phases by considering the mechanics in duplex systems[47–49]. Based on the above parameters, we propose a model to find the suitable deformation level for PSR in various eutectic systems containing soft and hard phases (see Methods and Supplementary Fig. 12). We further applied the PSR processing strategy to the industrial A357 casting aluminum alloy to prove its universality (Supplementary Fig. 13). The elongation significantly increases from ~8% in the AC state to ~23% after PSR. The tensile strength of the further strengthened A357 alloy is up to ~400 MPa, double that of the AC state. Theoretically and experimentally, it is convinced that PSR is a universal and powerful method to engineer the eutectic alloys as high-strength metallic materials.

In conclusion, we proposed a universal routine of PSR to ductilize eutectic alloys. The PSR, i.e., individual recrystallization of the soft phase and recovery of the hard phase, was achieved by tailoring the strain partitioning behavior between the two phases of eutectics. Compared with traditional processing methods for eutectic alloys, the PSR prevents premature fracture by eliminating the weak boundaries and thus triggers the intrinsically superior strain hardening capacity of the two phases through dislocation multiplication and potential phase transformation. In a typical FCC/B2 EHEA, ~35% uniform elongation and ~2 GPa true stress were achieved. The strain partitioning-dominated PSR mechanism ensures its good adaptability in other duplex alloys, as proven in the A357 alloy. This strategy will open new frontiers of eutectic alloys as high-strength metallic materials in model civilization by combining their excellent castability and strengthening capacity.

## Methods

### Materials preparation

The EHEAs with the nominal composition of $Ni_{30}Co_{30}Cr_{10}Fe_{10}Al_{18}W_2$ (at%) were prepared by vacuum arc melting in a Ti-gettered high-purity argon atmosphere. Ni, Co, Cr, Fe, Al, and W elements with at least 99.9 wt% purity were used as raw materials. Each ingot was re-melted 4 times to ensure the compositional homogeneity and then dropped into a water-cooled copper mold with a dimension of $80 \times 12 \times 5$ mm³ by gravity.

The AC alloy was denoted as the AC alloy. For the first cycle of PSR, the AC alloy was cold-rolled by 30% and annealed at 1200 °C for 20 mins (denoted as the PR alloy). For the second cycle of PSR, the PR alloy was cold-rolled by 30% and annealed at 1200 °C for 20 mins (denoted as the PSR alloy). For comparison, the PSR alloy was cold-rolled by 40% and annealed at 1200 °C for 20 mins to obtain the fully recrystallized alloy (denoted as the FR alloy). For further strengthening, the PSR alloy was cold-rolled by 30%, 60%, 75% and annealed at 700 °C for 3 hrs, 0.5 hrs, 0.5 hrs, respectively (denoted as the SPSR-1, SPSR-2, SPSR-3 alloys). The cooling method for all the annealing processes was water quenching.

To prove the universality of this method, a similar processing route was performed on the industrial casting A357 aluminum alloy ingot. For PSR, the AC A357 alloy was cold-rolled by 15% and annealed at 540 °C for 40 mins for two cycles, followed by 30% cold-rolled and annealed at 540 °C for 40 mins for two cycles. For further strengthening, the PSR A357 alloy was 80% cold-rolled and annealed at 160 °C for 2 hrs. The cooling method for all the annealing processes was water quenching.

### Mechanical tests

Quasi-static uniaxial tensile tests were performed on a TSMT EM6.504 tensile testing machine at room temperature with an initial strain rate

of $1 \times 10^{-3}\,\text{s}^{-1}$. Dog-bone-shaped specimens with a gauge length of 12.5 mm were fabricated by electro-discharge machining (EDM). A mechanical extensometer was used to monitor the strain. All tests were repeated at least three times to ensure the data reproducibility. Loading-unloading-reloading tests were conducted to calculate the back stress ($\sigma_b$) and effective stress ($\sigma_{eff}$)[50]. The samples were loaded at an initial strain rate of $1 \times 10^{-3}\,\text{s}^{-1}$ until reaching the designated strain, after which they were unloaded by the load-control mode to 20 N, followed by reloading at a strain rate of $1 \times 10^{-3}\,\text{s}^{-1}$ to the same applied strain before the next unloading. Nanoindentation tests for the PSR alloy were performed on a Hysitron TI980 triboindenter using a diamond Berkovich tip at room temperature. During each indentation, the load was linearly increased to 5000 μN within 5 s, then held for 2 s and unloaded within 5 s. Microstructure observation after indentation was conducted to determine the phase-specific indents. At least 5 points were tested for each phase to ensure the data reliability.

## Microstructural characterization

The microstructural analyses were conducted using a TESCAN MIRA3 field emission scanning electron microscope (FE-SEM) equipped with an OXFORD electron backscattered diffraction (EBSD) system. The specimens were mechanically polished, followed by electro-polishing for 5–10 s using the electrolyte of $HClO_4$ 10 vol% and $C_2H_5OH$ 90 vol% under a direct voltage of 30 V at room temperature. Channel 5 software was used to post-process the raw EBSD data.

In situ tensile experiments for the AC, FR, and PSR alloys were also conducted on the TESCAN MIRA3 FE-SEM. Tensile specimens with a gauge dimension of $3 \times 2 \times 1\,\text{mm}^3$ were fabricated by EDM. Prior to testing, the specimens were ground and finely polished by colloidal silica, followed by depositing a mono-layer of $SiO_2$ nanoparticles using the drop-casting technique[51]. During testing, the specimens were deformed at an initial strain rate of $1 \times 10^{-3}\,\text{s}^{-1}$ in a Kammrath & Weiss tensile stage. Secondary electron (SE) and in-beam SE images were taken for every 2% increase in nominal strain. For in situ μ-DIC analyses, the GOM Correlate software was used to process the images and calculate the equivalent von Mises strain, where facet size and overlap were optimized for better image quality in each experiment.

A JEM-2100F TEM was used to analyze the deformation mechanisms of the AC and PSR alloys. The specimens were extracted from the middle region of the gauge length, and two-beam conditions were used to image the dislocations. The TEM specimens were first cut by EDM and mechanically ground to about 60 μm. Then, they were punched into discs with a diameter of 3 mm and further thinned by ion-milling.

## Estimation of the strain partitioning during cold rolling

We assume that both the strain and stress of the constituent phases are proportional to their volume fractions by considering the mechanics in duplex system[47–49]:

$$\sigma_{EHEA} = \sigma_{FCC} V_{FCC} + \sigma_{B2} V_{B2}, \qquad (1)$$

$$\varepsilon_{EHEA} = \varepsilon_{FCC} V_{FCC} + \varepsilon_{B2} V_{B2}, \qquad (2)$$

where $\sigma_{FCC}$ and $\sigma_{B2}$ are the average true stress in the FCC and B2 phases, $\varepsilon_{FCC}$ and $\varepsilon_{B2}$ are the average true strain in the FCC and B2 phases, $V_{FCC}$ and $V_{B2}$ are the volume fractions of the FCC and B2 phases, respectively. The stress-strain relationships of the EHEA and its constituent FCC and B2 phases can be further expressed by the Swift equation[52]:

$$\sigma_{EHEA} = K_{EHEA} \left( \varepsilon_{0,EHEA} + \varepsilon_{EHEA} \right)^{n_{EHEA}}, \qquad (3)$$

$$\sigma_{FCC} = K_{FCC} \left( \varepsilon_{0,FCC} + \varepsilon_{FCC} \right)^{n_{FCC}}, \qquad (4)$$

$$\sigma_{B2} = K_{B2} \left( \varepsilon_{0,B2} + \varepsilon_{B2} \right)^{n_{B2}}, \qquad (5)$$

where $K_i$, $\varepsilon_{0,i}$, and $n_i$ ($i$ = EHEA, FCC, B2) are the proportional coefficients, constants related to elastic stress, and work hardening exponents of the EHEA and its constituent FCC and B2 phases, respectively. The parameters in Eqs. (3–5) can be obtained by fitting the stress-strain curves. Therefore, the average true strain and stress in the FCC and B2 phases during deformation can be calculated by combining Eqs. (1–5).

Considering the similar stress states between compression and cold-rolling[53], we employed the relatively simple compressive model to estimate the strain and stress partitioning behaviors. Nearly single-phase FCC and B2 alloys were fabricated by measuring the chemical compositions of the individual phases in the EHEA[54]. The AC FCC alloy was homogenized at 1200 °C for 2 hrs, cold-rolled 70%, and recrystallized at 1200 °C for 20 mins to obtain a grain size close to that of the EHEA. Then, the compressive engineering stress-strain curves of the EHEA, FCC, and B2 alloys were tested (Supplementary Fig. 12a). The true stress-strain curves are presented and fitted using the Swift equation in Supplementary Fig. 12b. To simplify, the elastic strain before yielding is ignored due to its small value and difficulty to be measured. The plastic strain of 0–30% is considered to avoid the nonuniform drum deformation at high strain. The fitted curves agree well with the experimental curves, proving the validity of the Swift equation in describing the work hardening behaviors of the EHEA, FCC, and B2 alloys. The fitted parameters of $K_i$, $\varepsilon_{0,i}$, and $n_i$ ($i$ = EHEA, FCC, B2) are summarized in Supplementary Table 1.

By combining Eqs. (1–5), the average true strain and stress in the FCC and B2 phases at different global strains are calculated and presented in Supplementary Fig. 12c, d. Generally, the average true strain and stress in both the FCC and B2 phases increase with the global strain, and the variations are similar to those of the dual-phase steels[55], confirming the reliability of the results. Besides, the proportion of the stress partitioned in the FCC phase increase gradually, indicating the more and more important role of FCC in strengthening due to the significant work hardening behavior. The calculated strains in the FCC and B2 phases at different global strains are summarized in Supplementary Table 2.

Based on the above understanding, we can summarize a universal equation to calculate the moderate deformation level for PSR. For eutectic alloys, the partitioned strain in the soft and hard phases under a given global strain ($\varepsilon$) can be described using the functions involving the intrinsic parameters of the eutectic alloy and its constituent soft and hard phases $K_i$, $\varepsilon_{0,i}$, $n_i$ ($i$ = eutectics, soft, hard), and the volume fraction of the soft phase $V_{soft}$:

$$\varepsilon_{soft} = f_1 \left( K_i, \varepsilon_{0,i}, n_i, V_{soft}, \varepsilon \right), \qquad (6)$$

$$\varepsilon_{hard} = f_2 \left( K_i, \varepsilon_{0,i}, n_i, V_{soft}, \varepsilon \right). \qquad (7)$$

To achieve PSR, the following conditions are recommended: $\varepsilon_{soft} > \varepsilon_{soft-rec}$ and $\varepsilon_{hard} < \varepsilon_{hard-rec}$, where $\varepsilon_{soft-rec}$ and $\varepsilon_{hard-rec}$ are the recrystallization critical strain for the soft and hard phases. Usually, the recrystallization critical strain of a given phase is in a finite range varied with annealing temperature. Therefore, the suitable deformation level for PSR can be recommended by combining the above inequalities. The stress state in cold-rolling is similar to that in the compression test[53], thus the strain partitioning behavior in the compression test can be taken as a reference.

The model indicates the universality of the PSR strategy in this study and provides practical guidance for application in various

**Article** https://doi.org/10.1038/s41467-022-32444-4

eutectic systems. It should be noted that this model is semi-quantitative by ignoring the information on microstructure and strain rate. Further modifications could be considered in the future. Considering that the parameter acquisition and the recrystallization critical strain of a given phase are complicated, it will also be a feasible way to directly characterize the PSR behaviors of the eutectic alloys under different rolling amounts and subsequent annealing.

## Data availability

The data that support the findings of this study are available from the corresponding author upon request.

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

## Acknowledgements
J.C.W. and F.H. acknowledge the financial support from the National Natural Science Foundation of China (Grant Nos. 51871183, 52001266). Z.J.W. acknowledges the financial support from the National Natural Science Foundation of Shaanxi Province in China (Grant No. 2020JQ-720). J.J.L. acknowledges the financial support from the Research Fund of the State Key Laboratory of Solidification Processing (NPU), China (Grant No. 2020-TS-06). H.S.K. acknowledges the support from the National Research Foundation of Korea (NRF) grant funded by the Korea government (MSIP) (Grant No. NRF-2021R1A2C3006662).

## Author contributions
Q.F.W. and Z.J.W. designed the research. Q.F.W. prepared the samples, and performed the microstructure characterization and mechanical tests. Q.F.W. and F.H. carried out the in situ tensile experiments. Q.F.W., F.H., and Z.J.W. analyzed the data. J.J.L., H.S.K., and J.C.W. discussed the results. Z.J.W., H.S.K., and J.C.W. supervised the research. Q.F.W., F.H., and Z.J.W. wrote the paper. All authors reviewed and contributed to the final manuscript.

## Competing interests
The authors declare no competing interests.

## Additional information
**Supplementary information** The online version contains

supplementary material available at https://doi.org/10.1038/s41467-022-32444-4.

