## [Peer Review File · Nature Communications]

Title: Phase-selective recrystallization makes eutectic high-entropy alloys ultra-ductileReviewers' comments:

Reviewer #1 (Remarks to the Author):

This is a very interesting paper and quite a novel approach to processing. I only have two comments:

1. it is not clear where the data in Figure 1c comes from. Is it from the references in the caption?
2. There could be more discussion of strain partitioning and recrystallization in EHEAs, both of which have been studied by others. The paper "Eutectic/Eutectoid Multi-Principle Component Alloys: A Review", I. Baker, M. Wu and Z. Wang, *Materials Characterization*, 147 (2019) 545-557. <https://doi.org/10.1016/j.matchar.2018.07.030> contains many references to papers on these topics that the authors can look up.

Reviewer #2 (Remarks to the Author):

The authors reported a novel HEA containing eutectic microstructure based on Ni₃₀Co₃₀Cr₁₀Fe₁₀Al₁₈W₂ composition and indicated that the alloy has a very high tensile strength and good ductility. The authors used a set of thermomechanical treatments to improve the ductility and strength of the alloy starting from an as-cast condition.

The topic of synergistic improvement of strength and ductility in HEAs is not new especially EHEAs.

Multiple recent publications have reported strategies to achieve this.

Such as *Materials Today* 41, 62-71 (2020); *Nat Commun* 10, 1-8 (2019); *Nature Communications* 10, no. 1 (2019): 1-10 etc; *Journal of Dynamic Behavior of Materials* 5, no. 1 (2019): 1-7 etc.

Moreover, the phase selective recrystallization or heterogeneous recrystallization has been shown to be effective in retaining ductility along with strengthening (*Proceedings of the National Academy of Sciences* 112, no. 47 (2015): 14501-14505, *Scripta Materialia* 156 (2018): 105-109, *Materials Research Letters* 6, no. 12 (2018): 676-682).

It is a classical metallurgical work and the results observed are in line with expectations and hence there is no appreciable novelty or ground-breaking finding that can excite a wide audience.

In view of this, the reviewer believes that it should be considered for publication in typical domain-oriented journals like *Acta Materialia*, *Materials Research Letters*, *MSEA* etc. However, there are significant shortcomings that need to be addressed before it can be published. Below are the details,

- 1) Basis of alloy composition selection has not been discussed.
- 2) As cast alloy conditions are most times, not the best performing condition for tensile deformation due to casting defects, porosity on the phase boundaries. Hence, HIP or cold rolling and recrystallization more often is performed to restore the ductility and premature failure of the material. Hence, the improvement of ductility from S1 to S3 may not be directly related to barely phase selective recrystallization. Does it mean if the alloy did not observe a phase selective recrystallization and both B2 and FCC recrystallized (probably due to higher percent cold rolling and prolonged high-temperature annealing) the alloy will not be ductilized?
- 3) The nanoparticles observed in B2 are not discussed. It seems there are two types of precipitates within B2. The structure of these precipitates was not established. Do they show any extra reflections in

SAED patterns? Are these precipitates present in all conditions?

Reviewer #3 (Remarks to the Author):

In this paper, the authors have presented a method (phase-selective recrystallization) to improve the ductility of the eutectic high-entropy alloy (EHEA) Ni₃₀Co₃₀Cr₁₀Fe₁₀Al₁₈W₂. It is also claimed that the proposed method is a universal way to achieve ultrastrong eutectic alloys. Despite the high quality of writing and analysis, I have the following concerns about the paper:

1- The novelty of this work is not clear, as there are studies in the literature that have discussed how the combination of heat treatment and cold-rolling increases the ductility of different high-entropy/ eutectic alloys. This idea was first presented by Bhattacharjee et al. [1], and it was followed by another paper [2] where they discussed how the size of the grain in the harder B2 phase is much smaller than the L12 phase. In both of these works, they have not specifically named it "phase-selective recrystallization" but they have investigated the difference in the grain size in the phases. Also in another study [3], it was shown how the ductility of the material has increased by the same method. Finally, in the current paper, the authors have attributed preferential recrystallization to the large misorientation gradient after cold rolling, but this concept has already been discussed in the two aforementioned papers [1,2].

2- While the authors have provided strong evidence that shows strain partitioning between the FCC and BCC phases, as well as the occurrence of recrystallization only in the FCC phase, they did not provide any evidence that shows the difference between strength of these two phases. Nanoindentation can be used for this purpose. The reason for demanding this is that the authors have claimed that the proposed method is universal for increasing the strength of eutectic alloys via phase-selective recrystallization, but the following questions come to mind: What if the strength of both phases are close to each other? And both undergo deformation and have enough driving force to experience recrystallization during further annealing. This is the case for AlCoCrFeNi₁ [2] and Al_{0.7}CoCrFeMnNi [4] eutectic high entropy alloys. If the as-cast alloy is not ductile enough to undergo sufficient deformation, then it is not possible to raise its ductility and strength via recrystallization because there is not enough driving force for recrystallization.

Based on the aforementioned reasons, I don't think the developed method is a universal method for eutectic alloys and can only be used for increasing the strength of eutectic alloys that satisfy these conditions: 1. the as-cast alloy must be ductile enough to undergo sufficient deformation before annealing, and 2. the difference between the strengths of phases must be significant enough.

[1] T. Bhattacharjee, I. S. Wani, S. Sheikh, I. T. Clark, T. Okawa, S. Guo, P. P. Bhattacharjee & N. Tsuji, Simultaneous Strength-Ductility Enhancement of a Nano-Lamellar AlCoCrFeNi_{2.1} Eutectic High Entropy Alloy by Cryo-Rolling and Annealing, Scientific Reports volume 8, Article number: 3276 (2018)

[2] I. Wani, T. Bhattacharjee, S. Sheikh, I. Clark, M. Park, T. Okawa, S. Guo, P.P. Bhattacharjee, N. Tsuji, Cold-rolling and recrystallization textures of a nano-lamellar AlCoCrFeNi_{2.1} eutectic high entropy alloy, Intermetallics 84 (2017) 42-51.

[3] I. Wani, T. Bhattacharjee, S. Sheikh, P.P. Bhattacharjee, S. Guo, N. Tsuji, Tailoring nanostructures and mechanical properties of AlCoCrFeNi_{2.1} eutectic high entropy alloy using thermo-mechanical processing, *Materials Science and Engineering*: 675 (2016) 99-109.

[4] H. Jeong, H. Park, W. Kim, Dynamic recrystallization and hot deformation mechanisms of a eutectic Al_{0.7}CoCrFeMnNi high-entropy alloy, *Journal of Alloys and Compounds* 871 (2021) 159488.

Response to reviewers' comments

Reviewer #1

Comment 1.1. This is a very interesting paper and quite a novel approach to processing. I only have two comments.

Response to comment 1.1:

Thank you very much for the positive evaluation. We appreciate your time and effort in reviewing our manuscript.

Comment 1.2. It is not clear where the data in Figure 1c comes from. Is it from the references in the caption?

Response to comment 1.2:

Thank you very much for the suggestion. Figure 1c in the original manuscript has been changed to Figure 1e in the revised manuscript. We have added the references in both the revised manuscript and figure caption.

“In Fig. 1e, we summarized the tensile properties of the present PSR EHEA and further strengthened PSR EHEAs together with AC, FR, and UFG EHEAs^{7,24,26-33}. The uniform elongation of the PSR EHEA (~33%) exceeds the maximum value (~25%) achieved by traditional thermomechanical treatments, while the engineering ultimate tensile strength approaches the strongest UFG EHEA. The further strengthened PSR EHEAs show great improvement in tensile strength compared with the traditional UFG in the previous reports^{7,24}.”

Fig. 1e, Ultimate tensile strength versus uniform elongation of the present PSR and further strengthened PSR EHEAs compared with traditional AC, FR, and UFG EHEAs^{7,24,26-33}.

Comment 1.3. There could be more discussion of strain partitioning and recrystallization in EHEAs, both of which have been studied by others. The paper “Eutectic/Eutectoid Multi-Principal Component Alloys: A Review” (I. Baker, M. Wu, Z. Wang, *Materials Characterization*, 147 (2019) 545-557) contains many references to papers on these topics that the authors can look up.

Response to comment 1.3:

Thank you very much for the suggestion and recommended reference. The recommended paper “Eutectic/Eutectoid Multi-Principal Component Alloys: A Review” summarizes the microstructures, mechanical properties, and thermomechanical processing of typical eutectic high-entropy alloys (EHEAs), which provides valuable information on the strain partitioning and recrystallization in EHEAs. We have cited this paper and the related papers to compare strain partitioning and recrystallization during phase-selective recrystallization and traditional full recrystallization.

“Traditional treatment of full recrystallization (FR) eliminates the lamellar structures, replaced by the equiaxed duplex structures with randomly distributed orientations (Fig. 1b). As reported, the lamellar structures can be retained in the FR alloys at very low annealing temperatures, resulting in the ultrafine-grained (UFG) EHEAs^{7,24}. The phase-selective recrystallization (PSR) brings a distinct microstructure compared with the AC and FR alloys. The FCC phase shows approximately equiaxed grains with random orientations, while the B2 phase exhibits a skeletal morphology with several specific orientations (Fig. 1c), i.e., phase-selective recrystallization. The increased frequency of twin boundaries in the FCC phase and low angle grain boundaries in the B2 phase confirm the separate recrystallization and recovery of the two phases (Fig. S1). After PSR, the interfaces between the FCC and B2 phases deviate from the original Kurdjumov-Sachs (K–S) orientation relationship²⁵, and the sizes of both the two phases increase slightly (Fig. S2).”

“The key of PSR originates from our critical thinking on tailoring the strain partitioning behaviors in duplex phase alloys: that is, moderate deformation will

induce two different levels of strain in the two phases, where one is for recrystallization while the other can only cause recovery during subsequent thermal treatment. Accordingly, the unique processing routes developed for PSR include two cycles of moderate deformation and annealing (Fig. 4a). After one cycle of thermomechanical treatment, the FCC phase in the irregular regions recrystallized partially (Fig. S10a-c). After two cycles of thermomechanical treatment, the FCC phase recrystallized completely while the B2 phase recovered and coarsened.

In view of the critical role of the strain partitioning between the two phases in PSR, it is necessary to be uncovered. Since the microhardness of the FCC phase (~ 4.44 GPa) is much lower than the B2 phase (~ 5.67 GPa) (Fig. S10d-f), it will bear more strain during deformation. In situ μ -DIC technique was used to further characterize the strain partitioning behaviors. A distinct strain difference exists between the two phases during tensile deformation (Fig. 4b). Quantitative analysis reveals that the local von Mises strain in the FCC phase is ~ 6 times higher than that in the B2 phase (Fig. 4c). Therefore, it is reasonable that cold rolling will induce more strain energy in the FCC phase, which promotes the subsequent individual recrystallization^{45,46}.”

Reviewer #2:

Comment 2.1. The authors reported a novel HEA containing eutectic microstructure based on $\text{Ni}_{30}\text{Co}_{30}\text{Cr}_{10}\text{Fe}_{10}\text{Al}_{18}\text{W}_2$ composition and indicated that the alloy has a very high tensile strength and good ductility. The authors used a set of thermomechanical treatments to improve the ductility and strength of the alloy starting from an as-cast condition. The topic of synergistic improvement of strength and ductility in HEAs is not new especially EHEAs. Multiple recent publications have reported strategies to achieve this. Such as Materials Today, 41 (2020) 62-71; Nature Communications, 10 (2019) 489; Nature Communications, 10 (2019) 5623; Journal of Dynamic Behavior of Materials, 5 (2019) 1-7.

Response to comment 2.1:

Thank you for the critical comments. However, it is unfair to ignore the innovation in the present paper by referring a general topic. **We know that the synergistic improvement of strength and ductility in high-entropy alloys (HEAs) and eutectic HEAs (EHEAs) has been extensively studied, but it does not mean that there will be no advances in this field. Instead, the present method is a totally new concept different from all the existing synergistic strengthening and ductilization in EHEAs.**

We have deeply compared the existing works to reveal our innovation. In the paper “Materials Today, 41 (2020) 62-71”, Shi et al. [1] fabricated an ultrafine-grained duplex $\text{Fe}_{20}\text{Co}_{20}\text{Ni}_{41}\text{Al}_{19}$ EHEA via cold rolling and low-temperature annealing. The highlight is the unique **multi-type twinning enabled sequential activation** of stress-dependent multiple hardening mechanism. They achieved an exceptional ductility of ~24%, but the ultimate tensile strength was only ~1.5 GPa.

In the paper “Nature Communications, 10 (2019) 489”, Shi et al. [2] achieved a higher ultimate tensile strength of ~1.6 GPa by engineering the ultrafine-grained duplex $\text{AlCoCrFeNi}_{2.1}$ EHEA. The **inherited lamellar morphology with nano-grains** is the highlight. The high strength can be attributed to the two-hierarchical constraint microstructure which provided effective back stress strengthening.

In the paper “Nature Communications, 10 (2019) 5623”, Ma et al. [3] summarized the wide variety of nanostructured heterogeneities in HEAs, including multicomponent

intermetallic nanoparticles, heterogeneous grain structures, dual phase eutectic lamellae, et al. For EHEAs, **there were only common literatures used strengthening and ductilization strategies.**

In the paper “Journal of Dynamic Behavior of Materials, 5 (2019) 1-7”, Gangireddy et al. [4] engineered precipitation-strengthened $Al_{0.7}CoCrFeNi$ near-eutectic HEA and discovered a **dynamic compressive strength of over 2 GPa under a strain rate of 2×10^3 m/s**. Since EHEAs exhibited significant strain rate sensitivity, the dynamic strength can be much higher than the quasi-static strength [5, 6].

At present, the 1.6 GPa ultimate tensile strength with 16% elongation is the best record for thermomechanically strengthened EHEAs with a standard sample size. It seems that it is impossible to make further advances in strengthening the EHEAs with these comprehensive reports of strength and ductility, and further strengthening and ductilization become extremely difficult because the strengthening methods have been exhausted.

Surprisingly, we found new mechanisms for further strengthening the EHEAs and proposed a brand-new phase-selective recrystallization (PSR) route for ultra-ductile or ultra-strong EHEAs. The 1.8 GPa ultimate tensile strength with 18% elongation, the 2.2 GPa ultimate tensile strength with 4% elongation, were achieved in EHEA for the first time, breaking the current record. Accordingly, the present work is completely different from traditional strategies both in mechanism and performance.

In the revised manuscript, we compared the tensile properties of the PSR EHEAs with traditional as-cast (AC), fully recrystallized (FR), ultrafine-grained (UFG) EHEAs. As shown in Fig. 1e, the PSR and further strengthened PSR EHEAs exhibited better mechanical properties, overcoming the strength-ductility trade-off.

“Accompanied by PSR, the ductility of EHEAs increases significantly. As shown in Fig. 1d, the true tensile stress-strain curves of the AC, FR, and PSR EHEAs do not show any differences in the yielding behavior but exhibit remarkable differences in the fracture stress and strain. The AC EHEA only has a fracture true strain of ~14%. The PSR EHEA with a well-tailored phase-selective recrystallization microstructure doubles the fracture strain to ~30%. In comparison, the classical FR EHEA only

increases the fracture strain slightly to ~17%. More importantly, the PSR EHEA exhibits a tremendous strain hardening capacity until fracture, leading to the high fracture true stress of ~1850 MPa, much higher than those of the AC and FR alloys, ~1390 and ~1460 MPa, respectively.

The excellent ductility and strain hardenability provide great potential to further improve the strength. By introducing dislocations and precipitations, the engineering ultimate tensile strength of the further strengthened PSR EHEAs was tuned between ~1.8 and 2.2 GPa (Fig. S3). Such ultrastrong eutectic alloys have never been reported to the authors' knowledge. In Fig. 1e, we summarized the tensile properties of the present PSR EHEA and further strengthened PSR EHEAs together with AC, FR, and UFG EHEAs^{7,24,26-33}. The uniform elongation of the PSR EHEA (~33%) exceeds the maximum value (~25%) achieved by traditional thermomechanical treatments, while the engineering ultimate tensile strength approaches the strongest UFG EHEA. The further strengthened PSR EHEAs show great improvement in tensile strength compared with the traditional UFG in previous reports^{7,24}.”

Fig. 1. d, Tensile true stress-strain curves of the AC, FR, and PSR EHEAs. **e.** Ultimate tensile strength versus uniform elongation of the present PSR and further strengthened PSR EHEAs^{7,24,26-33} compared with traditional AC, FR, and UFG EHEAs^{7,24,26-33}.

References for Response to comment 2.1:

- [1] P. Shi et al., Multistage work hardening assisted by multi-type twinning in ultrafine-grained heterostructural eutectic high-entropy alloys. *Materials Today* 41, 62-71 (2020).
- [2] P. Shi et al., Enhanced strength-ductility synergy in ultrafine-grained eutectic high-entropy alloys by inheriting microstructural lamellae. *Nature Communications* 10, 489 (2019).

- [3] E. Ma, X. Wu, Tailoring heterogeneities in high-entropy alloys to promote strength–ductility synergy. *Nature Communications* 10, 5623 (2019).
- [4] S. Gangireddy, B. Gwalani, R. Banerjee, R. S. Mishra, High strain rate response of $Al_{0.7}CoCrFeNi$ high entropy alloy: dynamic strength over 2 GPa from thermomechanical processing and hierarchical microstructure. *Journal of Dynamic Behavior of Materials* 5, 1-7 (2019).
- [5] B. Gwalani et al., Influence of ordered L12 precipitation on strain-rate dependent mechanical behavior in a eutectic high entropy alloy. *Scientific Reports* 9, 6371 (2019).
- [6] S. Muskeri, V. Hasannaemi, R. Salloom, M. Sadeghilaridjani, S. Mukherjee, Small-scale mechanical behavior of a eutectic high entropy alloy. *Scientific Reports* 10, 2669 (2020).

Comment 2.2. Moreover, the phase selective recrystallization or heterogeneous recrystallization has been shown to be effective in retaining ductility along with strengthening (Proceedings of the National Academy of Sciences, 112(47) (2015): 14501-14505; *Materials Research Letters*, 6(12) (2018) 676-682); *Scripta Materialia*, 156 (2018) 105-109. It is a classical metallurgical work and the results observed are in line with expectations and hence there is no appreciable novelty or ground-breaking finding that can excite a wide audience.

Response to comment 2.2:

Thank you for the critical comments. **It should be noted that there are fundamental differences between PSR in the present study and traditional heterogeneous recrystallization with partially recrystallized microstructures.**

In the paper “Proceedings of the National Academy of Sciences, 112(47) (2015)”, Wu et al. [7] reported a **partially recrystallized** Ti alloy with the soft micro-grained lamellae embedded in the hard ultrafine-grained lamella matrix. The heterogeneous lamella structure unites ultrafine-grain strength with coarse-grain ductility due to significant back stress hardening and dislocation hardening.

In the paper “*Materials Research Letters*, 6(12) (2018) 676-682”, Shukla et al. [8] combined **heterogeneous partial recrystallized grain structure**, B2 precipitates and nanotwins in the FCC-based $Al_{0.3}CoCrFeNi$ HEA. During deformation, the strain partitioning between fine and coarse grains produced long-range back stress and thus led to the simultaneous enhancement of strength and ductility.

As for the paper “*Scripta Materialia*, 156 (2018) 105-109”, Shukla et al. [9]

reported a hierarchy EHEA with four distinct microstructural features: (i) deformed lamellar FCC region; (ii) deformed lamellar BCC region with precipitates; (iii) recrystallized FCC region with precipitates; (iv) recrystallized BCC region. They attributed the enhanced properties to the **combination of recrystallized region and non-recrystallized lamellae region** of the FCC and BCC phases.

It is clear that the current heterogeneous recrystallization designs are based on partial recrystallization, irrelevant to the PSR in the present work. The PSR in this study is a brand-new concept with recrystallized soft phase and recovered hard phase, which can be reflected in the following two aspects.

On the one hand, the PSR process results in a **fully recrystallized soft phase embedded in the skeleton of a recovered hard phase in dual-phase materials**, which has never been discovered. During the unique processing routes developed for PSR, the moderate deformation will induce two different levels of strain in the two phases, where one is for recrystallization while the other can only cause recovery during subsequent thermal treatment. This mechanism also ensures the universality of the processing routes for dual-phase materials containing both soft and hard phases.

On the one hand, the **ductilization mechanism of PSR is completely different** from traditional heterogeneous recrystallization. In the PSR EHEA, our brand-new strategy primarily focuses on eliminating and confining the crack nucleation during deformation by tuning the strain partitioning of eutectic alloys. With reduced and confined early cracks, the excellent strain hardening capacity of both phases in eutectics were completely released, rendering a twofold tensile elongation compared to those of AC and FR EHEAs.

We have added additional experiments to clarify the innovations and reorganized the manuscript to highlight the two unique characteristics of PSR in the revised manuscript.

“We present phase-selective recrystallization in a model EHEA $\text{Ni}_{30}\text{Co}_{30}\text{Cr}_{10}\text{Fe}_{10}\text{Al}_{18}\text{W}_2$ (in atomic percentage) consisting of face-centered cubic (FCC) and ordered body-centered cubic (B2) phases^{22,23} in the current work. In the as-cast (AC) state, the EHEA exhibits a typical eutectic-dendritic structure, where the dendrite stems are well-aligned lamellar structures while the interlamellar regions are irregular duplex structures (Fig. 1a). Traditional treatment of full recrystallization (FR)

eliminates the lamellar structures, replaced by the equiaxed duplex structures with randomly distributed orientations (Fig. 1b). As reported, the lamellar structures can be retained in the FR alloys at very low annealing temperatures, resulting in the ultrafine-grained (UFG) EHEAs^{7,24}. The phase-selective recrystallization (PSR) brings a distinct microstructure compared with the AC and FR alloys. The FCC phase shows approximately equiaxed grains with random orientations, while the B2 phase exhibits a skeletal morphology with several specific orientations (Fig. 1c), i.e., phase-selective recrystallization. The increased frequency of twin boundaries in the FCC phase and low angle grain boundaries in the B2 phase confirm the separate recrystallization and recovery of the two phases (Fig. S1). After PSR, the interfaces between the FCC and B2 phases deviate from the original Kurdjumov-Sachs (K-S) orientation relationship²⁵, and the sizes of both the two phases increase slightly (Fig. S2).”

Fig. 1. Microstructures and mechanical properties of the PSR EHEA. a-c, Electron backscattering diffraction (EBSD) inverse pole figure (IPF) maps of the FCC (upper row) and B2 (lower row) phases in the AC, FR, and PSR EHEAs, respectively. The insets show the corresponding pole figure (PF) maps. d, Tensile true stress-strain curves of the AC, FR, and PSR EHEAs. e, Ultimate tensile strength versus uniform elongation of the present PSR and further strengthened PSR EHEAs compared with traditional AC, FR, and UFG EHEAs^{7,24,26-33}.

The dramatic increase in ductility of the PSR EHEA calls for a deeper understanding of the relationship between microstructures and deformation mechanisms. We firstly examined the strain hardening rate curves to understand the

deformation behaviors at the macroscopic level (Fig. 2a). The PSR alloy exhibits a three-stage strain hardening behavior, i.e., the curve drops rapidly after yielding (Stage 1), then becomes stable gradually (Stage 2), and finally decreases slowly (Stage 3) until reaching the tensile plastic instability criterion³⁴. However, the AC and FR alloys only have a two-stage strain hardening behavior and fracture before meeting the instability criterion. The premature failures of the AC and FR alloys are responsible for their relatively low ductility. In comparison, the PSR alloy exhausts its strain hardening capacity before failure, which is unusual in eutectic alloys.

We attribute these distinct strain hardening behaviors of the AC, FR, and PSR EHEAs to their entirely different microstructures (Figs. 1a-c) based on the close examination of the fracture and deformation mechanisms. Figures 2b-d present the fracture cross-sections and surfaces of the AC, FR, PSR EHEAs during in-situ tensile tests to distinguish the different fracture mechanisms. On the cross-section of the fractured AC specimen, a large number of microcracks appear at the phase boundaries (PBs) in the lamellar regions, which have an angle of $\sim 45^\circ$ with the tensile direction (Figs. 2b and S4). The connection of these microcracks results in the zig-zag morphologies of the main crack and partially flat fracture surface. Therefore, the premature failure of the AC EHEA is caused by the decohesion of the lamellar PBs with specific orientations. After eliminating the lamellar structures by recrystallization, the primary crack mainly originates from the GBs of the B2 phase (indicated by the yellow arrows), and the fracture surfaces exhibit typical dimple morphologies in the FR and PSR EHEAs (Figs. 2c and d). As revealed by the in situ tensile tests (Figs. S5 and S6), the microcracks occur at the GBs of B2 at $\sim 17\%$ nominal strain, which is consistent with the fracture strain of the FR alloy, further proving its GB cracks-induced premature failure. However, the PSR alloy shows amazing crack resistance. The microcrack density of the fractured PSR specimen is only one-third of the FR specimen, even under a twofold nominal strain (Fig. 2d). This is because the B2 phase in PSR is an integrated skeleton while the B2 phase in FR is recrystallized with GBs. The GB density of B2 in the PSR EHEA is much lower than that in the FR EHEA due to the non-recrystallization state after PSR. Therefore, from the perspective of fracture, the PSR prevents premature failures and enhances ductility by eliminating the lamellar structures in the AC alloy while avoiding the high-density PBs of the B2 phase in the FR alloy.

We conducted in situ microscopic-digital image correlation (μ -DIC) experiments to reveal the underlying reasons for the different fracture modes. In the as-cast state, the grains having 45° oriented lamellar structures with the tensile direction exhibit severe strain localization (Fig. 2e) and become crack initiations. Such an orientation-dependent strain localization behavior has also been observed in lamellar-structured TiAl alloys³⁵ and pearlitic steels³⁶, and can be well interpreted by the dislocation pile-up model. After PSR, the grain-scale strain localization is released, and strain partitioning only exists between the two phases of FCC and B2 (Fig. 3f), thereby preventing the crack nucleation^{37,38}. In this case, the weak GBs of the load-bearing B2 phase become the crack initiations in the FR and PSR alloys.

Fig. 2. Fracture mechanisms of the PSR EHEA. **a**, Strain hardening rate curves of the AC, FR, and PSR EHEAs. **b-d**, Fracture cross-sections and surfaces of the AC, FR, and PSR EHEAs. The microcracks within the B2 phase are marked by the yellow arrows. The inset in **c** shows the EBSD IPF map of the B2 phase around a typical microcrack, revealing the crack initiation at the GB. The inset in **d** compares the number density of microcracks between FR and PSR EHEAs, revealing the decreased cracking propensity of the PSR EHEA. **e and f**, In situ microscopic-digital image correlation (μ -DIC) results of the AC and PSR EHEAs during tensile deformation,

respectively, revealing the severe strain localization in the AC EHEA and uniform strain distribution in the PSR EHEA. The tensile direction is horizontal in all images.

Apart from these distinct fracture mechanisms, the PSR also stimulated extra deformation mechanisms, which assure the third stage of strain hardening in the PSR EHEA (Fig. 2a). To validate this standpoint, we conducted detailed transmission electron microscope (TEM) analyses. As shown in Fig. 3a, dense plane slip bands are characterized in the FCC phase of the fractured AC alloy, while much fewer dislocations are detected in the B2 phase with a nominal strain 15%. Moreover, dislocations in the B2 phase tend to appear near the PBs. In the PSR alloy with a similar nominal strain (18%, Fig. 3b), however, both the FCC and B2 phases exhibit a high density of dislocations. We also examined the dislocation structures of the PSR EHEA at nominal strains of 0.5% and 8%, respectively (Fig. S7). Well-developed planar slip bands in both phases show that the FCC and B2 phases in the PSR EHEA contribute to the strain hardening simultaneously from the onset of plastic deformation. As the strain further increases, the dislocation densities significantly increase in both phases (Fig. 3c), illustrating their superior dislocation storage capacity. Especially for the B2 phase, even though no evidence for slip transfer across the PBs^{39,40} is detected, the naturally superior deformability allows a high density of dislocations to be activated. As a result, the geometrically necessary dislocations significantly multiplied due to the heterogeneous microstructure, promoting the sustainable increase of back stress strengthening^{41,42}. By conducting the loading-reloading-loading experiments, the back stress is quantitatively measured to be ~900 MPa near failure (Fig. S8), accounting for the ultrahigh strain hardening capacity of the PSR EHEA.

More importantly, a phase transformation in the B2 phase is activated by the increased strain. We observed a B2 \rightarrow body-centered tetragonal (BCT) phase transformation in the PSR EHEA during deformation. The selected area electron diffraction (SAED) patterns of the B2 phase taken from the [001] zone axis transfer from square-arranged spots before deformation to parallelogram-arranged spots after deformation (Fig. 3d), indicating the distinct interplanar spacings between (200) and (020) planes. Quantitative calculations (Figs. 3d-f and S9) reveal the crystal structure expands ~6% along the (200) direction while compresses ~5% along the (002) and

(020) directions after deformation, as schematically illustrated in Fig. 3g. The ductile-transformable B2 phase⁴³, has been proven to enhance the fatigue life significantly in the duplex HEA⁴⁴. In the present PSR EHEA, the phase transformation enhances the deformability of the B2 phase and promotes the coordinated deformation between the soft and hard phases. Therefore, the sustainable strain hardening of the PSR EHEA beyond 18% strain is accomplished by activating high-density dislocations in the FCC and B2 phases (Fig. 3c) and phase transformation in the B2 phase (Fig. 3g).

Fig. 3. Deformation mechanisms of the PSR EHEA. **a**, Dislocation substructures in the fractured AC EHEA. **b and c**, Dislocation substructures in the 18% and 35% stretched PSR EHEAs, respectively. **d-f**, Selected area electron diffraction (SAED) patterns of the original B2 phase in the fractured PSR specimen taken from the zone axis of [001], [011], $[\bar{1}11]$, respectively, revealing the body-centered tetragonal (BCT) crystal structure. The superlattices are marked by the red circles. **g**, Schematic diagram illustrating the crystal structure transformation before and after deformation

References for Response to comment 2.2:

- [7] X. Wu et al., Heterogeneous lamella structure unites ultrafine-grain strength with coarse-grain ductility. *Proceedings of the National Academy of Sciences* 112, 14501-14505 (2015).
- [8] S. Shukla et al., Hierarchical features infused heterogeneous grain structure for extraordinary strength-ductility synergy. *Materials Research Letters* 6, 676-682 (2018).
- [9] S. Shukla, T. Wang, S. Cotton, R. S. Mishra, Hierarchical microstructure for improved fatigue properties in a eutectic high entropy alloy. *Scripta Materialia* 156, 105-109 (2018).

Comment 2.3. Basis of alloy composition selection has not been discussed.

Response to comment 2.3:

Thank you very much for the suggestion. The PSR method is universal for eutectic alloys. We did not choose the alloy composition deliberately, only selected a model EHEA to illustrate the significant effect of PSR and further strengthening on dual-phase alloys with soft and hard phases. The EHEA composition has been reported in our published papers [10, 11]. We have cited relevant references in the revised manuscript. “We present phase-selective recrystallization in a model EHEA $\text{Ni}_{30}\text{Co}_{30}\text{Cr}_{10}\text{Fe}_{10}\text{Al}_{18}\text{W}_2$ (in atomic percentage) consisting of face-centered cubic (FCC) and ordered body-centered cubic (B2) phases^{22,23} in the current work.”

In the supplemental materials, the industrial A357 casting aluminum alloy was also used to identify the validity of the strategy. “We further applied the PSR processing strategy to the industrial A357 casting aluminum alloy for further illustration (Fig. S11). The elongation significantly increases from ~8% in the as-cast state to ~23% after PSR. The tensile strength of the further strengthened A357 alloy is up to ~400 MPa, double that of the as-cast state. Theoretically and experimentally, it is convinced that PSR is a universal and powerful method to engineer the eutectic alloys as super strong metallic materials.”

Fig. S11. Engineering stress-strain curves of the AC, PSR and further strengthened SPSR A357 casting aluminum alloys. The AC A357 alloy exhibits poor ductility of ~8%. After phase-selective recrystallization, the ductility of the PSR A357 alloy increases to ~23%. After further strengthening by introducing dislocations and precipitations, the SPSR A357 alloy exhibits a high tensile strength of ~400 MPa, double that of the as-cast state. These results prove the versatility of phase-selective recrystallization and further strengthening routes for various eutectic systems.

References for Response to comment 2.3:

- [10] Q. Wu et al., A casting eutectic high entropy alloy with superior strength-ductility combination. *Materials Letters* 253, 268-271 (2019).
- [11] Q. Wu et al., Uncovering the eutectics design by machine learning in the Al-Co-Cr-Fe-Ni high entropy system. *Acta Materialia* 182, 278-286 (2020).

Comment 2.4. As cast alloy conditions are most times, not the best performing condition for tensile deformation due to casting defects, porosity on the phase boundaries. Hence, HIP or cold rolling and recrystallization more often is performed to restore the ductility and premature failure of the material. Hence, the improvement of ductility from S1 to S3 may not be directly related to barely phase selective recrystallization. Does it mean if the alloy did not observe a phase selective recrystallization and both B2 and FCC recrystallized (probably due to higher percent cold rolling and prolonged high-temperature annealing) the alloy will not be ductilized?

Response to comment 2 4:

Thank you very much for the suggestion. Generally, the traditional thermomechanical treatment can increase the ductility by reducing the casting defects, porosity on the phase boundaries. Then we have added the fully recrystallized (FR) EHEA as a benchmark to show the contribution of PSR in the revised manuscript based on your suggestion. The results revealed that the FR EHEA with equiaxed duplex structures exhibited an increased tensile elongation compared with the as-cast (AC) EHEA (from ~15% to ~18%). However, both of them are much lower than the phase-selectively recrystallized (PSR) EHEA (~35%), as shown in Fig 1d.

In the revision, “We present phase-selective recrystallization in a model EHEA $\text{Ni}_{30}\text{Co}_{30}\text{Cr}_{10}\text{Fe}_{10}\text{Al}_{18}\text{W}_2$ (in atomic percentage) consisting of face-centered cubic (FCC) and ordered body-centered cubic (B2) phases^{22,23} in the current work. In the as-cast (AC) state, the EHEA exhibits a typical eutectic-dendritic structure, where the dendrite stems are well-aligned lamellar structures while the interlamellar regions are irregular duplex structures (Fig. 1a). Traditional treatment of full recrystallization (FR) eliminates the lamellar structures, replaced by the equiaxed duplex structures with randomly distributed orientations (Fig. 1b). As reported, the lamellar structures can be retained in the FR alloys at very low annealing temperatures, resulting in the ultrafine-grained (UFG) EHEAs^{7,24}. The phase-selective recrystallization (PSR) brings a distinct microstructure compared with the AC and FR alloys. The FCC phase shows approximately equiaxed grains with random orientations, while the B2 phase exhibits a skeletal morphology with several specific orientations (Fig. 1c), i.e., phase-selective recrystallization. The increased frequency of twin boundaries in the FCC phase and low angle grain boundaries in the B2 phase confirm the separate recrystallization and recovery of the two phases (Fig. S1). After PSR, the interfaces between the FCC and B2 phases deviate from the original Kurdjumov-Sachs (K–S) orientation relationship²⁵, and the sizes of both the two phases increase slightly (Fig. S2).”

Fig. 1. Microstructures and mechanical properties of the PSR EHEA. a-c, Electron backscattering diffraction (EBSD) inverse pole figure (IPF) maps of the FCC (upper row) and B2 (lower row) phases in the AC, FR, and PSR EHEAs, respectively. The insets show the corresponding pole figure (PF) maps. d, Tensile true stress-strain curves of the AC, FR, and PSR EHEAs. e, Ultimate tensile strength versus uniform elongation of the present PSR and further strengthened PSR EHEAs compared with traditional AC, FR, and UFG EHEAs^{7,24,26-33}.

In the revision, we further analyzed the ductilization mechanism and revealed that the FR EHEA eliminated the weak lamellar phase boundaries in the AC alloy but introduced new weak grain boundaries of the B2 phase, as shown in Fig. 2. Only the

PSR EHEA without weak phase and grain boundaries can fully trigger the strain hardening capacity of both the FCC and B2 phases and lead to the high fracture true stress of ~1850 MPa. “The dramatic increase in ductility of the PSR EHEA calls for a deeper understanding of the relationship between microstructures and deformation mechanisms. We firstly examined the strain hardening rate curves to understand the deformation behaviors at the macroscopic level (Fig. 2a). The PSR alloy exhibits a three-stage strain hardening behavior, i.e., the curve drops rapidly after yielding (Stage 1), then becomes stable gradually (Stage 2), and finally decreases slowly (Stage 3) until reaching the tensile plastic instability criterion³⁴. However, the AC and FR alloys only have a two-stage strain hardening behavior and fracture before meeting the instability criterion. The premature failures of the AC and FR alloys are responsible for their relatively low ductility. In comparison, the PSR alloy exhausts its strain hardening capacity before fracture, which is unusual in eutectic alloys.

We attribute these distinct strain hardening behaviors of the AC, FR, and PSR EHEAs to their entirely different microstructures (Figs. 1a-c) based on the close examination of the fracture and deformation mechanisms. Figures 2b-d present the fracture cross-sections and surfaces of the AC, FR, PSR EHEAs during in-situ tensile tests to distinguish the different fracture mechanisms. On the cross-section of the fractured AC specimen, a large number of microcracks appear at the phase boundaries (PBs) in the lamellar regions, which have an angle of ~45° with the tensile direction (Figs. 2b and S4). The connection of these microcracks results in the zig-zag morphologies of the main crack and partially flat fracture surface. Therefore, the premature failure of the AC EHEA is caused by the decohesion of the lamellar PBs with specific orientations. After eliminating the lamellar structures by recrystallization, the primary crack mainly originates from the GBs of the B2 phase (indicated by the yellow arrows), and the fracture surfaces exhibit typical dimple morphologies in the FR and PSR EHEAs (Figs. 2c and d). As revealed by the in situ tensile tests (Figs. S5 and S6), the microcracks occur at the GBs of B2 at ~17% nominal strain, which is consistent with the fracture strain of the FR alloy, further proving its GB cracks-induced premature failure. However, the PSR alloy shows amazing crack resistance. The microcrack density of the fractured PSR specimen is only one-third of the FR specimen, even under a twofold nominal strain (Fig. 2d). This is because the B2 phase in PSR is an integrated skeleton while the B2 phase in FR is recrystallized

with GBs. The GB density of B2 in the PSR EHEA is much lower than that in the FR EHEA due to the non-recrystallization state after PSR. Therefore, from the perspective of fracture, the PSR prevents premature failures and enhances ductility by eliminating the lamellar structures in the AC alloy while avoiding the high-density PBs of the B2 phase in the FR alloy.

We conducted in situ microscopic-digital image correlation (μ -DIC) experiments to reveal the underlying reasons for the different fracture modes. In the as-cast state, the grains having 45° oriented lamellar structures with the tensile direction exhibit severe strain localization (Fig. 2e) and become crack initiations. Such an orientation-dependent strain localization behavior has also been observed in lamellar-structured TiAl alloys³⁵ and pearlitic steels³⁶, and can be well interpreted by the dislocation pile-up model. After PSR, the grain-scale strain localization is released, and strain partitioning only exists between the two phases of FCC and B2 (Fig. 3f), thereby preventing the crack nucleation^{37,38}. In this case, the weak GBs of the load-bearing B2 phase become the crack initiations in the FR and PSR alloys.”

Fig. 2. Fracture mechanisms of the PSR EHEA. a, Strain hardening rate curves of the AC, FR, and PSR EHEAs. b-d, Fracture cross-sections and surfaces of the AC, FR,

and PSR EHEAs. The microcracks within the B2 phase are marked by the yellow arrows. The inset in **c** shows the EBSD IPF map of the B2 phase around a typical microcrack, revealing the crack initiation at the GB. The inset in **d** compares the number density of microcracks between FR and PSR EHEAs, revealing the decreased cracking propensity of the PSR EHEA. **e and f**, In situ microscopic-digital image correlation (μ -DIC) results of the AC and PSR EHEAs during tensile deformation, respectively, revealing the severe strain localization in the AC EHEA and uniform strain distribution in the PSR EHEA. The tensile direction is horizontal in all images.

Comment 2.5. The nanoparticles observed in B2 are not discussed. It seems there are two types of precipitates within B2. The structure of these precipitates was not established. Do they show any extra reflections in SAED patterns? Are these precipitates present in all conditions?

Response to comment 2.5:

Thank you very much for the suggestion. According to the high-resolution transmission electron microscope images and corresponding fast Fourier transform patterns shown in Fig R1, the lath-shaped precipitates rich in Co, Cr, Fe elements can be confirmed as the FCC phase having K-S orientation with the B2 matrix, while the rod-shaped precipitates rich in W elements can be confirmed as the μ phase. The precipitates only exist within the B2 matrix in the further strengthened PSR EHEA. In the AC, FR, and PSR EHEAs, no precipitates were detected both in the FCC and B2 matrix.

Fig. R1. TEM characterizations of the further strengthened PSR EHEA. **a**, Bright field image showing the dislocations and precipitates. The insets show selected area electron diffraction (SAED) patterns of the FCC and B2 matrix. **b**, EDS mappings taken from the B2 matrix showing the elemental distribution of the precipitates. **c and d**, HRTEM images and corresponding FFT images of the lath-shaped and rod-shaped precipitates, revealing the crystal structures of FCC and μ , respectively.

In the revised manuscript, we focused on the ductilization mechanism induced by PSR to highlight the innovations, and removed the detailed results about the precipitates in the further strengthened PSR EHEAs. The title of the paper was also changed from “Tuning eutectic alloys to ultrastrong metallic materials via phase-selective recrystallization” to “Phase-selective recrystallization makes eutectic high-entropy alloys ultra-ductile”.

Reviewer #3:

Comment 3.1. In this paper, the authors have presented a method (phase-selective recrystallization) to improve the ductility of the eutectic high-entropy alloy (EHEA) $\text{Ni}_{30}\text{Co}_{30}\text{Cr}_{10}\text{Fe}_{10}\text{Al}_{18}\text{W}_2$. It is also claimed that the proposed method is a universal way to achieve ultrastrong eutectic alloys. Despite the high quality of writing and analysis, I have the following concerns about the paper.

The novelty of this work is not clear, as there are studies in the literature that have discussed how the combination of heat treatment and cold-rolling increases the ductility of different high-entropy eutectic alloys. This idea was first presented by Bhattacharjee et al. (Scientific Reports, 8 (2018) 3276), and it was followed by another paper (Intermetallics, 84 (2017) 42-51) where they discussed how the size of the grain in the harder B2 phase is much smaller than the L12 phase. In both of these works, they have not specifically named it "phase-selective recrystallization" but they have investigated the difference in the grain size in the phases. Also in another study (Materials Science and Engineering A, 675 (2016) 99-109), it was shown how the ductility of the material has increased by the same method. Finally, in the current paper, the authors have attributed preferential recrystallization to the large misorientation gradient after cold rolling, but this concept has already been discussed in the two aforementioned papers (Scientific Reports, 8 (2018) 3276; Intermetallics, 84 (2017) 42-51).

Response to comment 3.1:

Thank you for the positive comments on our writing and analysis. We are appreciated for your suggestions to clarify the novelty. **The PSR is apparently different from the traditional thermomechanical treatments in the recommended literature, from the processing, obtained microstructures and mechanical behaviors. And hence, the related mechanisms are different.**

Firstly, the PSR has a different routine compared with traditional thermomechanical treatments. The traditional thermomechanical treatments were firstly introduced to tailoring the mechanical properties of EHEAs by Bhattacharjee et al. [12, 13]. Bhattacharjee et al. [12, 13] obtained fully recrystallized (FR) microstructures with different grain sizes through severe plastic deformation and annealing at different temperatures. **However, in this study, cycles**

of moderate deformation and annealing routes were utilized for PSR. The processing route of phase selective recrystallization (PSR) proposed in this study is entirely different from traditional thermomechanical treatments.

“The key of PSR originates from our critical thinking on tailoring the strain partitioning behaviors in duplex phase alloys: that is, moderate deformation will induce two different levels of strain partition in the two phases, where one is for recrystallization while the other can only cause recovery during subsequent thermal treatment. Accordingly, the unique processing routes developed for PSR include two cycles of moderate deformation and annealing (Fig. 4a). After one cycle of thermomechanical treatment, the FCC phase in the irregular regions recrystallized partially (Fig. S10a-c). After two cycles of thermomechanical treatment, the FCC phase recrystallized completely while the B2 phase recovered and coarsened.

In view of the critical role of the strain partitioning between the two phases in PSR, it is necessary to be uncovered. Since the microhardness of the FCC phase (~4.44 GPa) is much lower than the B2 phase (~5.67 GPa) (Fig. S10d-f), it will bear more strain during deformation. In situ μ -DIC technique was used to further characterize the strain partitioning behaviors. A distinct strain difference exists between the two phases during tensile deformation (Fig. 4b). Quantitative analysis reveals that the local von Mises strain in the FCC phase is ~6 times higher than that in the B2 phase (Fig. 4c). Therefore, it is reasonable that cold rolling will induce more strain energy in the FCC phase than B2 phase, which promotes the subsequent individual recrystallization^{45,46} in FCC phase but only recovery in B2 phase.”

Fig. 4. Processing routes and formation mechanisms of the phase-selective recrystallization. **a**, Schematic illustration of the processing routes and corresponding microstructure evolution. For PSR, the AC EHEA was cold-rolled 30% and annealed at 1200 °C for two cycles. **b**, In situ μ -DIC results of the AC EHEA during the tensile test, revealing the strain partitioning between the FCC and B2 phases during deformation. **c**, Variation of the local von Mises strain along the yellow arrow in **b**, revealing the much higher strain partitioned to the FCC phase.

Fig. S10. a and b, SEM images confirming the phase-specific indentations of FCC and B2 in the PSR EHEA. **c**, Load-displacement curves for the FCC and B2 phases.

Secondly, the microstructure of PSR EHEA is entirely different from FR

EHEAs. Bhattacharjee et al. [12, 13] obtained FR EHEA with regular duplex microstructures where the **recrystallized FCC phase can be regarded as matrix and the recrystallized B2 particles were isolated distributed within it.** It is reasonable that the isolated B2 phase with a lower volume fraction exhibited a finer grain size compared with the FCC matrix. However, in this study, the PSR microstructure exhibited **a fully recrystallized soft FCC phase embedded in the skeleton of a hard B2 phase.** We have compared the microstructures of FR and PSR EHEAs in the revised manuscript.

“In the as-cast (AC) state, the EHEA exhibits a typical eutectic-dendritic structure, where the dendrite stems are well-aligned lamellar structures while the interlamellar regions are irregular duplex structures (Fig. 1a). Traditional treatment of full recrystallization (FR) eliminates the lamellar structures, replaced by the equiaxed duplex structures with randomly distributed orientations (Fig. 1b). As reported, the lamellar structures can be retained in the FR alloys at very low annealing temperatures, resulting in the ultrafine-grained (UFG) EHEAs^{7,24}. The phase-selective recrystallization (PSR) brings a distinct microstructure compared with the AC and FR alloys. The FCC phase shows approximately equiaxed grains with random orientations, while the B2 phase exhibits a skeletal morphology with several specific orientations (Fig. 1c), i.e., phase-selective recrystallization. The increased frequency of twin boundaries in the FCC phase and low angle grain boundaries in the B2 phase confirm the separate recrystallization and recovery of the two phases (Fig. S1). After PSR, the interfaces between the FCC and B2 phases deviate from the original Kurdjumov-Sachs (K-S) orientation relationship²⁵, and the sizes of both the two phases increase slightly (Fig. S2).”

Fig. 1. Microstructures and mechanical properties of the PSR EHEA. a-c, Electron backscattering diffraction (EBSD) inverse pole figure (IPF) maps of the FCC (upper row) and B2 (lower row) phases in the AC, FR, and PSR EHEAs, respectively. The insets show the corresponding pole figure (PF) maps.

Thirdly, the mechanical properties of PSR EHEA are much better than the FR EHEAs, breaking the record of eutectic alloys over 2 GPa tensile strength with a regular sample size. Bhattacharjee et al. [12] improved the uniform elongation of $\text{AlCoCrFeNi}_{2.1}$ EHEA from ~15% to ~23% by recrystallization. However, the strain hardening capacity was severely reduced due to the FR microstructures. As a result, the ultimate tensile strength was lower than 1.2 GPa. The PSR EHEA in this study exhibited a much higher uniform elongation of ~35%, surpassing all FR EHEAs. Besides, the ultrahigh ductility and work hardening capacity can be utilized for obtaining ultrahigh strength over 2 GPa in EHEAs with a regular sample size, which has never been reported. We have compared the tensile properties of FR and PSR EHEAs in the revised manuscript.

“Accompanied by PSR, the ductility of EHEAs increases significantly. As shown in Fig. 1d, the true tensile stress-strain curves of the AC, FR, and PSR EHEAs do not show any differences in the yielding behavior but exhibit remarkable differences in

the fracture stress and strain. The AC EHEA only has a fracture true strain of ~14%. The PSR EHEA with a well-tailored phase-selective recrystallization microstructure doubles the fracture strain to ~30%. In comparison, the classical FR EHEA only increases the fracture strain slightly to ~17%. More importantly, the PSR EHEA exhibits a tremendous strain hardening capacity until fracture, leading to the high fracture true stress of ~1850 MPa, much higher than those of the AC and FR alloys, ~1390 and ~1460 MPa, respectively.

The excellent ductility and strain hardenability provide great potential to further improve the strength. By introducing dislocations and precipitations, the engineering ultimate tensile strength of the further strengthened PSR EHEAs was tuned between ~1.8 and 2.2 GPa (Fig. S3). Such ultrastrong eutectic alloys have never been reported to the authors' knowledge. In Fig. 1e, we summarized the tensile properties of the present PSR EHEA and further strengthened PSR EHEAs together with AC, FR, and UFG EHEAs^{7,24,26-33}. The uniform elongation of the PSR EHEA (~33%) exceeds the maximum value (~25%) achieved by traditional thermomechanical treatments, while the engineering ultimate tensile strength approaches the strongest UFG EHEA. The further strengthened PSR EHEAs show great improvement in tensile strength compared with the traditional UFG in previous reports^{7,24}.

Fig. 1. d, Tensile true stress-strain curves of the AC, FR, and PSR EHEAs. **e.** Ultimate tensile strength versus uniform elongation of the present PSR and further strengthened PSR EHEAs compared with traditional AC, FR, and UFG EHEAs^{7,24,26-33}.

Finally, although it is well-known that the preferential recrystallization occurs with the large misorientation gradient after cold rolling, the present study

is the first attempt to utilize it to obtain fully recrystallized FCC phase together with recovered B2 phase, an unprecedented PSR microstructure with superior mechanical properties. We have carefully checked the recommended references. Bhattacharjee et al. [14] investigated the partially recrystallized microstructure of the AlCoCrFeNi_{2.1} EHEA after **heavily deformed by cryo-rolling** and annealing. They attributed the preferential recrystallization of the coarse non-lamellar region to the large misorientation gradient induced by deformation. Their results and analysis are reasonable. In this study, the preferential recrystallization of the FCC phase after **moderately deformed by cold-rolling** and annealing can also be attributed to the large stored strain and misorientation gradient. As for the other recommended literature, “Intermetallics, 84 (2017) 42-51”, there is no discussion about the preferential recrystallization of regions with the large misorientation gradient after cold rolling.

We have cited relevant references when discussing the mechanism of PSR in the revised manuscript. “Since the microhardness of the FCC phase (~4.44 GPa) is much lower than the B2 phase (~5.67 GPa) (Fig. S10d-f), it will bear more strain during deformation. In situ μ -DIC technique was used to further characterize the strain partitioning behaviors. A distinct strain difference exists between the two phases during tensile deformation (Fig. 4b). Quantitative analysis reveals that the local von Mises strain in the FCC phase is ~6 times higher than that in the B2 phase (Fig. 4c). Therefore, it is reasonable that cold rolling will induce more strain energy in the FCC phase, which promotes the subsequent individual recrystallization^{45,46}.”

Based on the above, we believe that the PSR strategy in this study has significant innovations in processing, microstructures, and properties, which achieves great breakthroughs in the field of EHEAs.

References for Response to comment 3.1:

- [12] I. S. Wani et al., Tailoring nanostructures and mechanical properties of AlCoCrFeNi 2.1 eutectic high entropy alloy using thermo-mechanical processing. *Materials Science and Engineering: A* 675, 99-109 (2016).
- [13] I. S. Wani et al., Cold-rolling and recrystallization textures of a nano-lamellar AlCoCrFeNi_{2.1} eutectic high entropy alloy. *Intermetallics* 84, 42-51 (2017).
- [14] T. Bhattacharjee et al., Simultaneous strength-ductility enhancement of a nano-lamellar

Comment 3.2. While the authors have provided strong evidence that shows strain partitioning between the FCC and BCC phases, as well as the occurrence of recrystallization only in the FCC phase, they did not provide any evidence that shows the difference between strength of these two phases. Nanoindentation can be used for this purpose. The reason for demanding this is that the authors have claimed that the proposed method is universal for increasing the strength of eutectic alloys via phase-selective recrystallization.

Response to comment 3.2:

Thank you very much for the suggestions. **The strength difference between the FCC and B2 phases is indeed a critical factor affecting the PSR.** We have conducted the nanoindentation test to illustrate the strength difference in the revised manuscript. As revealed by Fig. S10, the FCC phase exhibited a lower microhardness of ~4.44 GPa than the B2 phase of ~5.67 GPa. Combined with the in situ microscopic-digital image correlation (μ -DIC) results in Fig. 4 where a distinct strain difference exists between the two phases during tensile deformation can be determined, it is reasonable that cold rolling will induce more strain energy in the FCC phase, which promotes the subsequent individual recrystallization.

In the revision, “The key of PSR originates from our critical thinking on tailoring the strain partitioning behaviors in duplex phase alloys: that is, moderate deformation will induce two different levels of strain partition in the two phases, where one is for recrystallization while the other can only cause recovery during subsequent thermal treatment. Accordingly, the unique processing routes developed for PSR include two cycles of moderate deformation and annealing (Fig. 4a). After one cycle of thermomechanical treatment, the FCC phase in the irregular regions recrystallized partially (Fig. S10a-c). After two cycles of thermomechanical treatment, the FCC phase recrystallized completely while the B2 phase recovered and coarsened.

In view of the critical role of the strain partitioning between the two phases in PSR, it is necessary to be uncovered. Since the microhardness of the FCC phase (~4.44 GPa) is much lower than the B2 phase (~5.67 GPa) (Fig. S10d-f), it will bear more strain during deformation. In situ μ -DIC technique was used to further

characterize the strain partitioning behaviors. A distinct strain difference exists between the two phases during tensile deformation (Fig. 4b). Quantitative analysis reveals that the local von Mises strain in the FCC phase is ~ 6 times higher than that in the B2 phase (Fig. 4c). Therefore, it is reasonable that cold rolling will induce more strain energy in the FCC phase than B2 phase, which promotes the subsequent individual recrystallization^{45,46} in FCC phase but only recovery in B2 phase.”

Fig. S10. a and b, SEM images confirming the phase-specific indentations of FCC and B2 in the PSR EHEA. c, Load-displacement curves for the FCC and B2 phases.

Fig. 4. Processing routes and formation mechanisms of the phase-selective recrystallization. a, Schematic illustration of the processing routes and corresponding microstructure evolution. For PSR, the AC EHEA was cold-rolled 30%

and annealed at 1200 °C for two cycles. **b**, In situ μ -DIC results of the AC EHEA during the tensile test, revealing the strain partitioning between the FCC and B2 phases during deformation. **c**, Variation of the local von Mises strain along the yellow arrow in **b**, revealing the much higher strain partitioned to the FCC phase.

Comment 3.3. The following questions come to mind: What if the strengths of both phases are close to each other? And both undergo deformation and have enough driving force to experience recrystallization during further annealing. This is the case for AlCoCrFeNi_{2.1} (Intermetallics, 84 (2017) 42-51) and Al_{0.7}CoCrFeMnNi (Journal of Alloys and Compounds, 871 (2021) 159488) eutectic high entropy alloys. If the as-cast alloy is not ductile enough to undergo sufficient deformation, then it is not possible to raise its ductility and strength via recrystallization because there is not enough driving force for recrystallization. Based on the aforementioned reasons, I don't think the developed method is a universal method for eutectic alloys and can only be used for increasing the strength of eutectic alloys that satisfy these conditions: 1. the as-cast alloy must be ductile enough to undergo sufficient deformation before annealing, and 2. the difference between the strengths of phases must be significant enough.

Response to comment 3.3:

Thank you very much for the suggestions and comments. **The PSR process is universal and can be applied in most eutectic alloys for the following reasons.**

Firstly, there is almost no eutectics in which the strengths of both phases are close to each other. Generally, from the perspective of eutectic formation, eutectic alloys are composed of two phases with distinct compositions and even crystal structure [15, 16]. Then commonly, the two phases in eutectics have hardness differences. For example, even in the Pb-Sn eutectic system, both of the phases are soft but still have distinguishable hardness differences. The key for achieving PSR is to find the suitable deformation content before annealing. Moreover, the constituent phases in the referred AlCoCrFeNi_{2.1} [13] and Al_{0.7}CoCrFeMnNi [17] EHEAs have been pointed out to have significant hardness differences. For the AlCoCrFeNi_{2.1} EHEA, the hardness of FCC and B2 phases was measured to be 4.03 and 4.91 HV by nanoindentation tests [18]. For the Al_{0.7}CoCrFeMnNi EHEA, Jeong et al. [17] pointed

out that “the BCC phase and FCC phase have different hardness, high local stress and localized deformation can be created at interfaces” in the recommended literature.

Secondly, this method can also be used to the as-cast alloy even if it is not ductile enough. In the supplementary materials, the A357 alloy was tested for this case. As shown in Fig. S11, the elongation significantly increases from ~8% in the as-cast state to ~23% after PSR. For dual-phase alloys with poor plasticity, we can apply relatively low moderate deformation to achieve recrystallization in the soft phase gradually. To achieve PSR, the as-cast A357 alloy was cold-rolled by 15% and annealed for two cycles, with a further 30% cold-rolled and annealed for two cycles. The key issue is to find the suitable deformation content providing driving force for recrystallization and annealing conditions. Therefore, the present PSR process can improve the ductility of eutectic alloys with limited tensile ductility. However, for the very brittle eutectic alloys, the method may be invalid.

Fig. S11. Engineering stress-strain curves of the AC, PSR and further strengthened SPSR A357 casting aluminum alloys. The AC A357 alloy exhibits poor ductility of ~8%. After phase-selective recrystallization, the ductility of the PSR A357 alloy increases to ~23%. After further strengthening by introducing dislocations and precipitations, the SPSR A357 alloy exhibits a high tensile strength of ~400 MPa, double that of the as-cast state. These results prove the versatility of phase-selective recrystallization and further strengthening routes for various eutectic systems.

In the revised manuscript, we have an outlook to emphasize the applicable conditions for PSR in dual-phase alloys based on the suggestions and comments.

“This concept is versatile for various duplex alloys with soft and hard phases and

opens new frontiers for traditional eutectic alloys as ultra-strong metallic materials.” “Based on this understanding, we can design the moderate deformation level to achieve the PSR in various eutectic systems containing soft and hard phases.” “In conclusion, we proposed a universal routine of PSR to ductilize eutectic alloys. The PSR, i.e., individual recrystallization of the soft phase and recovery of the hard phase, was achieved by tailoring the strain partitioning behavior between the two phases of eutectics.”

References for Response to comment 3.2:

- [15] G. A. Chadwick, Eutectic alloy solidification. *Progress in Materials Science* 12, 99-182 (1963).
- [16] R. Elliott, *Eutectic solidification processing: crystalline and glassy alloys*. (Elsevier, 2013).
- [17] H. T. Jeong, H. K. Park, W. J. Kim, Dynamic recrystallization and hot deformation mechanisms of a eutectic $\text{Al}_{0.7}\text{CoCrFeMnNi}$ high-entropy alloy. *Journal of Alloys and Compounds* 871, 159488 (2021).
- [18] S. Muskeri, V. Hasannaemi, R. Salloom, M. Sadeghilaridjani, S. Mukherjee, Small-scale mechanical behavior of a eutectic high entropy alloy. *Scientific Reports* 10, 2669 (2020).

REVIEWER COMMENTS

Reviewer #2 (Remarks to the Author):

The authors have responded to the review comments well and have provided additional results to suffice the novelty of the work. I feel comfortable now recommending the paper for publication.

Reviewer #3 (Remarks to the Author):

In the revision the authors answered most the raised questions, but one main concern remains as they still claim that the proposed method is the universal way to achieve ultra-strong eutectic alloys. The authors in their response stated: “the key for achieving PSR is to find the suitable deformation content before annealing” and reported 30% cold rolling. But they did not explain how they came up with this suitable deformation content for the investigated high entropy alloy. Is there any formulation for calculating suitable deformation? Or this is a try and error procedure? Furthermore, they did not report in the supplementary the amount of suitable deformation for A357 alloy for achieving PSR. Since authors have claimed the proposed method is the universal procedure for achieving PSR in eutectic alloys, they need to present a universal relation for calculating suitable deformation before annealing for all the eutectic alloys. One can guess the relation need to be a function of strength of the two phases of eutectic alloy, the difference between their strength, and total strength and ductility (should be considered to prevent crack formation in the simple) of alloy.

Response to reviewers' comments

Reviewer #2

Comment 2.1. The authors have responded to the review comments well and have provided additional results to suffice the novelty of the work. I feel comfortable now recommending the paper for publication.

Response to Comment 2.1:

Thank you very much for your positive comments, we appreciated your time and efforts in reviewing our manuscript.

Reviewer #3

Comment 3.1. In the revision the authors answered most the raised questions, but one main concern remains as they still claim that the proposed method is the universal way to achieve ultrastrong eutectic alloys.

Response to Comment 3.1:

Thank you very much for your positive evaluation and valuable comments, we appreciated your time and efforts in reviewing our manuscript. We have provided more experiments and conducted more discussion to address the universality of the phase-selective recrystallization (PSR) strategy in the revised manuscript. All changes have been highlighted in yellow in the revised manuscript.

In the revisions,

“Theoretically, the partitioned strain in the constituent phases during deformation can be described as the functions related to the strength and work hardening parameters of the eutectic alloy and constituent phases, and volume fraction of the constituent phases by considering the mechanics in dual-phase systems⁴⁷⁻⁴⁹. Based on the above parameters, we proposed a model to find the suitable deformation level for PSR in various eutectic systems containing soft and hard phases (see Methods and Supplementary Fig. 12).”

“For eutectic alloys, the partitioned strain in the soft and hard phases under a given global strain (ϵ) can be described using the functions involving the intrinsic parameters of the eutectic alloy and its constituent soft and hard phases $K_i, \epsilon_{0,i}, n_i$ ($i = eutectics, soft, hard$), and the volume fraction of the soft phase V_{soft} :

$$\varepsilon_{soft} = f_1(K_i, \varepsilon_{0,i}, n_i, V_{soft}, \varepsilon), \quad (6)$$

$$\varepsilon_{hard} = f_2(K_i, \varepsilon_{0,i}, n_i, V_{soft}, \varepsilon). \quad (7)$$

To achieve PSR, the following conditions are recommended: $\varepsilon_{soft} > \varepsilon_{soft-rec}$ and $\varepsilon_{hard} < \varepsilon_{hard-rec}$, where $\varepsilon_{soft-rec}$ and $\varepsilon_{hard-rec}$ are the recrystallization critical strain for the soft and hard phases. Usually, the recrystallization critical strain of a given phase is in a finite range varied with annealing temperature. Therefore, the suitable deformation level for PSR can be recommended by combining the above inequalities.”

Comment 3.2. The authors in their response stated: “the key for achieving PSR is to find the suitable deformation content before annealing” and reported 30% cold rolling. But they did not explain how they came up with this suitable deformation content for the investigated high entropy alloy. Is there any formulation for calculating suitable deformation? Or this is a try and error procedure? Furthermore, they did not report in the supplementary the amount of suitable deformation for A357 alloy for achieving PSR.

Response to Comment 3.2:

Thank you very much for your valuable comments and suggestions. The suitable deformation amount for PSR of the EHEA was determined by experiments in the very beginning. For this purpose, the as-cast EHEA was cold-rolled 15%, 30%, 45%, and 60% and annealed at 1200 °C for 20 mins. Then, the recovery and recrystallization behaviors of the constituent FCC and B2 phases were characterized, as shown in Supplementary Fig. 10. For the 15%-rolled and annealed alloy, neither the FCC nor B2 phases recrystallized, but only recovered. For the 30%-rolled and annealed alloy, the FCC phase recrystallized partially, while the B2 phase recovered. For the 45%-rolled and annealed alloy, the B2 phase began to recrystallize. Accordingly, the suitable deformation amount of 30% was chosen for the following PSR.

In the revisions, we have added the experimental results and provided a schematic to show the recovery and recrystallization behaviors of the FCC and B2 phases under different deformation amounts and subsequent annealing in Supplementary Fig. 10.

“The recovery and recrystallization behaviors of the FCC and B2 phases in the EHEA under different deformation amounts and subsequent annealing were

investigated experimentally in Supplementary Fig. 10. Apparently, 30% is a suitable deformation amount to achieve partial recrystallization in FCC while only recovery in B2. Accordingly, the unique processing routes developed for PSR include two cycles of 30% moderate deformation and annealing (Fig. 4a).”

Supplementary Fig. 10. EBSD IPF maps of the FCC (upper row) and B2 (lower row) phases of the AC EHEA after rolling for different deformation amounts and annealed at 1200 °C for 20 mins. a, 15%. b, 30%. c, 45%. d, 60%. For the 15%-rolled and annealed alloy, neither the FCC nor B2 phases recrystallized, but only recovered. For the 30%-rolled and annealed alloy, the FCC phase recrystallized partially, while the B2 phase recovered. For the 45%-rolled and annealed alloy, the recrystallization volume fraction of FCC increased, and the B2 phase began to recrystallize. For the 60%-rolled and annealed alloy, the FCC phase recrystallized completely, while the B2 phase recrystallized partially. e, Schematic showing the recovery and recrystallization behaviors of the FCC and B2 phases under different deformation amounts and subsequent annealing.

Furthermore, the suitable deformation amount of the A357 alloy for achieving PSR was also determined by experiments and provided in the Methods in the revised manuscript. “For phase-selective recrystallization, the as-cast A357 alloy was cold-rolled by 15% and annealed at 540 °C for 40 mins for two cycles, followed by 30% cold-rolled and annealed at 540 °C for 40 mins for two cycles.”

Comment 3.3. Since authors have claimed the proposed method is the universal procedure for achieving PSR in eutectic alloys, they need to present a universal relation for calculating suitable deformation before annealing for all the eutectic alloys. One can guess the relation need to be a function of strength of the two phases of eutectic alloy, the difference between their strength, and total strength and ductility (should be considered to prevent crack formation in the simple) of alloy.

Response to Comment 3.3:

Thank you very much for these inspired comments and suggestions. In the revisions, a framework was proposed to evaluate the suitable deformation level before annealing for PSR. We derived a universal criterion to calculate the suitable deformation level for various eutectic alloys with soft and hard phases after reviewing the theoretical models about the strain partitioning in dual-phase alloys.

In the model, it assumes that both the strain and stress of the constituent phases are proportional to their volume fractions by considering the mechanics in dual-phase system¹⁻³:

$$\sigma_{eutectic} = \sigma_{soft}V_{soft} + \sigma_{hard}V_{hard}, \quad (1)$$

$$\varepsilon_{eutectic} = \varepsilon_{soft}V_{soft} + \varepsilon_{hard}V_{hard}, \quad (2)$$

where σ_{soft} and σ_{hard} are the average true stress in the soft and hard phases, ε_{soft} and ε_{hard} are the average true strain in the soft and hard phases, V_{soft} and V_{hard} are the volume fractions of the soft and hard phases, respectively. The stress-strain relationships of the eutectic alloy and its constituent soft and hard phases can be further expressed by the Swift equation⁴:

$$\sigma_{eutectic} = K_{eutectic}(\varepsilon_{0,eutectic} + \varepsilon_{eutectic})^{n_{eutectic}}, \quad (3)$$

$$\sigma_{soft} = K_{soft}(\varepsilon_{0,soft} + \varepsilon_{soft})^{n_{soft}}, \quad (4)$$

$$\sigma_{hard} = K_{hard}(\varepsilon_{0,hard} + \varepsilon_{hard})^{n_{hard}}, \quad (5)$$

where K_i , $\varepsilon_{0,i}$, and n_i ($i = eutectic, soft, hard$) are the proportional coefficients, constants related to elastic stress, and work hardening exponents of the eutectic alloy and its constituent soft and hard phases, respectively. The parameters in Eqs. (3)-(5) can be obtained by fitting the stress-strain curves.

The average true strain and stress in the soft and hard phases under a given global strain (ε) can be described as functions related to the parameters by combining Eqs. (1)-(5):

$$\varepsilon_{soft} = f_1(K_i, \varepsilon_{0,i}, n_i, V_{soft}, \varepsilon), \quad (6)$$

$$\varepsilon_{hard} = f_2(K_i, \varepsilon_{0,i}, n_i, V_{soft}, \varepsilon). \quad (7)$$

To achieve PSR, the following conditions are recommended: $\varepsilon_{soft} > \varepsilon_{soft-rec}$ and $\varepsilon_{hard} < \varepsilon_{hard-rec}$, where $\varepsilon_{soft-rec}$ and $\varepsilon_{hard-rec}$ are the recrystallization critical strain for the soft and hard phases, respectively. In this case, the universal relation to calculate the suitable deformation level for PSR in various eutectic alloys can be obtained by combining the above inequalities. The parameters K_i , $\varepsilon_{0,i}$, and n_i represent the strength and work hardening behavior of the constituent phases, and V_{soft} represents the microstructural information.

We further applied the model in the present EHEA, as shown in Methods, Supplementary Fig. 12, and Supplementary Tables 1 and 2 in the revised manuscript. Considering the similar stress states between compression and cold-rolling⁵, we employed the relatively simple compressive model to estimate the strain and stress partitioning behaviors. Nearly single-phase FCC and B2 alloys were fabricated by measuring the chemical compositions of the individual phases in the EHEA⁶, and corresponding compressive curves were tested and fitted to obtain the parameters. The predictions of the model are shown in Supplementary Fig. 12. The variations of the average strain and stress in the FCC and B2 phases are similar to those of the dual-phase steels⁷, confirming the reliability of the model. The model can not only prove the universality of the PSR strategy but also provide practical guidance when applied in various eutectic systems. Considering the parameter acquisition is complicated, and it will also be a feasible way to directly characterize the PSR behaviors of the eutectic alloys under different rolling amounts and subsequent annealing.

Supplementary Fig. 12. Estimation of the partitioned strain and stress in the FCC and B2 phases at different global strains. **a**, Engineering stress-strain curves of the FCC, B2, and EHEA alloys. **b**, Experimental true stress-strain curves of the FCC, B2, and EHEA alloys with only the plastic strain of 0-30% considered, and corresponding fitting curves. **c**, Calculated average true strain in the FCC and B2 phases at different global strains. **d**, Calculated average true stress in the FCC and B2 phases at different global strains.

Supplementary Table 1. Fitted parameters of K_i , $\varepsilon_{0,i}$, and n_i ($i = E, FCC, B2$) in

Supplementary Fig. 12b.

Alloy	K_i , MPa	$\varepsilon_{0,i}$	n_i
EHEA	2311.81 ± 1.36	0.0059 ± 0.000076	0.2894 ± 0.000388
FCC	1956.46 ± 0.43	0.0157 ± 0.000039	0.4058 ± 0.000178
B2	3294.30 ± 1.47	0.0358 ± 0.000170	0.3023 ± 0.000435

Supplementary Table 2. Calculated strain in the FCC and B2 phases at different global strains (%).

Global engineering strain	Global true strain	Average true strain in FCC	Average engineering strain in FCC	Average true strain in B2	Average engineering strain in B2
15	16	17	16	15	14
30	36	46	37	21	19
45	60	83	56	25	22

In the revisions,

“Theoretically, the partitioned strain in the constituent phases during deformation can be described as the functions related to the strength and work hardening parameters of the eutectic alloy and constituent phases, and volume fraction of the constituent phases by considering the mechanics in dual-phase systems⁴⁷⁻⁴⁹. Based on the above parameters, we proposed a model to find the suitable deformation level for PSR in various eutectic systems containing soft and hard phases (see Methods and Supplementary Fig. 12).”

“Estimation of the partitioned strain during cold rolling

We assume that both the strain and stress of the constituent phases are proportional to their volume fractions by considering the mechanics in dual-phase system⁴⁷⁻⁴⁹:

$$\sigma_{EHEA} = \sigma_{FCC}V_{FCC} + \sigma_{B2}V_{B2}, \quad (1)$$

$$\varepsilon_{EHEA} = \varepsilon_{FCC}V_{FCC} + \varepsilon_{B2}V_{B2}, \quad (2)$$

where σ_{FCC} and σ_{B2} are the average true stress in the FCC and B2 phases, ε_{FCC} and ε_{B2} are the average true strain in the FCC and B2 phases, V_{FCC} and V_{B2} are the volume fractions of the FCC and B2 phases, respectively. The stress-strain relationships of the EHEA and its constituent FCC and B2 phases can be further expressed by the Swift equation⁵²:

$$\sigma_{EHEA} = K_{EHEA}(\varepsilon_{0,EHEA} + \varepsilon_{EHEA})^{n_{EHEA}}, \quad (3)$$

$$\sigma_{FCC} = K_{FCC}(\varepsilon_{0,FCC} + \varepsilon_{FCC})^{n_{FCC}}, \quad (4)$$

$$\sigma_{B2} = K_{B2}(\varepsilon_{0,B2} + \varepsilon_{B2})^{n_{B2}}, \quad (5)$$

where K_i , $\varepsilon_{0,i}$, and n_i ($i = EHEA, FCC, B2$) are the proportional coefficients, constants related to elastic stress, and work hardening exponents of the EHEA and its constituent FCC and B2 phases, respectively. The parameters in Eqs. (3)-(5) can be obtained by fitting the stress-strain curves. Therefore, the average true strain and stress in the FCC and B2 phases during deformation can be calculated by combining Eqs. (1)-(5).

Considering the similar stress states between compression and cold-rolling⁵³, we employed the relatively simple compressive model to estimate the strain and stress partitioning behaviors. Nearly single-phase FCC and B2 alloys were fabricated by measuring the chemical compositions of the individual phases in the EHEA⁵⁴. The as-cast FCC alloy was homogenized at 1200 °C for 2 hrs, cold-rolled 70%, and recrystallized at 1200 °C for 20 mins to obtain a grain size close to that of the EHEA. Then, the compressive engineering stress-strain curves of the EHEA, FCC, and B2 alloys were tested (Supplementary Fig. 12a). The true stress-strain curves were presented and fitted using the Swift equation in Supplementary Fig. 12b. To simplify, the elastic strain before yielding was ignored due to its small value and difficulty to be measured. The plastic strain of 0-30% was considered to avoid the nonuniform drum deformation at high strain. The fitted curves agree well with the experimental curves, proving the validity of the Swift equation in describing the work hardening behaviors of the EHEA, FCC, and B2 alloys. The fitted parameters of K_i , $\varepsilon_{0,i}$, and n_i ($i = EHEA, FCC, B2$) were summarized in Supplementary Table 1.

By combining Eqs. (1)-(5), the average true strain and stress in the FCC and B2 phases at different global strains were calculated and presented in Supplementary Figs. 12c and d. Generally, the average true strain and stress in both the FCC and B2 phases increased with the global strain, and the variations are similar to those of the dual-phase steels⁵⁴, confirming the reliability of the results. Besides, the proportion of the stress partitioned in the FCC phase increased gradually, indicating the more and more important role of FCC in strengthening due to the significant work hardening behavior. The calculated strains in the FCC and B2 phases at different global strains were summarized in Supplementary Table 2.

Based on the above understanding, we can summarize a universal equation to

calculate the moderate deformation level for PSR. For eutectic alloys, the partitioned strain in the soft and hard phases under a given global strain (ε) can be described using the functions involving the intrinsic parameters of the eutectic alloy and its constituent soft and hard phases $K_i, \varepsilon_{0,i}, n_i$ ($i = eutectics, soft, hard$), and the volume fraction of the soft phase V_{soft} :

$$\varepsilon_{soft} = f_1(K_i, \varepsilon_{0,i}, n_i, V_{soft}, \varepsilon), \quad (6)$$

$$\varepsilon_{hard} = f_2(K_i, \varepsilon_{0,i}, n_i, V_{soft}, \varepsilon). \quad (7)$$

To achieve PSR, the following conditions are recommended: $\varepsilon_{soft} > \varepsilon_{soft-rec}$ and $\varepsilon_{hard} < \varepsilon_{hard-rec}$, where $\varepsilon_{soft-rec}$ and $\varepsilon_{hard-rec}$ are the recrystallization critical strain for the soft and hard phases. Usually, the recrystallization critical strain of a given phase is in a finite range varied with annealing temperature. Therefore, the suitable deformation level for PSR can be recommended by combining the above inequalities. The stress state in cold-rolling is similar to that in the compression test⁵³, thus the strain partition behavior in the compression test can be taken as a reference.

The model indicates the universality of the PSR strategy in this study and provides practical guidance for application in various eutectic systems. It should be noted that this model is semi-quantitative by ignoring the information on microstructure and strain rate. Further modifications could be considered in the future. Considering that the parameter acquisition and the recrystallization critical strain of a given phase are complicated, it will also be a feasible way to directly characterize the PSR behaviors of the eutectic alloys under different rolling amounts and subsequent annealing.”

References for Response to Comment 3.3

1. Tamura, I., Tomota, Y. & Ozawa, M. Strength and ductility of Fe-Ni-C alloys composed of austenite and martensite with various strength. The 3rd International Conference on Strength of Metals and Alloys (1973).
2. Cho, K. & Gurland, J. The law of mixtures applied to the plastic deformation of two-phase alloys of coarse microstructures. Metallurgical Transactions A 19, 2027-2040 (1988).
3. Lian, J., Jiang, Z. & Liu, J. Theoretical model for the tensile work hardening behaviour of dual-phase steel. Materials Science and Engineering: A 147, 55-65

(1991).

4. Swift, H. W. Plastic instability under plane stress. *Journal of the Mechanics and Physics of Solids* 1, 1-18 (1952).
5. Dieter, G. E. *Mechanical Metallurgy*, SI Metric Edition, London. (1988).
6. Wu, Q. et al. Rapid alloy design from superior eutectic high-entropy alloys. *Scripta Materialia* 219, 114875 (2022).
7. Su, Y. L. & Gurland, J. Strain partition, uniform elongation and fracture strain in dual-phase steels. *Materials Science and Engineering* 95, 151-165 (1987).

REVIEWERS' COMMENTS

Reviewer #3 (Remarks to the Author):

The authors have provided satisfactory responses, and the changes in the manuscript have addressed the raised concerns.